de Mendoza *et al. Genome Biology*     (2022) 23:163

# Large-scale manipulation of promoter DNA methylation reveals context-specific transcriptional responses and stability

Alex de Mendoza[1,2,3*†], Trung Viet Nguyen[1,2†], Ethan Ford[1,2†], Daniel Poppe[1,2], Sam Buckberry[1,2], Jahnvi Pflueger[1,2], Matthew R. Grimmer[4,5,6], Sabine Stolzenburg[7], Ozren Bogdanovic[1,8,9], Alicia Oshlack[10,11], Peggy J. Farnham[4], Pilar Blancafort[2,7,12] and Ryan Lister[1,2*]

†Alex de Mendoza, Trung Viet Nguyen and Ethan Ford contributed equally to this work.

*Correspondence: a.demendozasoler@qmul.ac.uk; ryan.lister@uwa.edu.au

[2] Harry Perkins Institute of Medical Research, QEII Medical Centre and Centre for Medical Research, The University of Western Australia, Perth, WA 6009, Australia
[3] School of Biological and Behavioural Sciences, Queen Mary University of London, Mile End Road, London E1 4NS, UK
Full list of author information is available at the end of the article

## Abstract

**Background:** Cytosine DNA methylation is widely described as a transcriptional repressive mark with the capacity to silence promoters. Epigenome engineering techniques enable direct testing of the effect of induced DNA methylation on endogenous promoters; however, the downstream effects have not yet been comprehensively assessed.

**Results:** Here, we simultaneously induce methylation at thousands of promoters in human cells using an engineered zinc finger-DNMT3A fusion protein, enabling us to test the effect of forced DNA methylation upon transcription, chromatin accessibility, histone modifications, and DNA methylation persistence after the removal of the fusion protein. We find that transcriptional responses to DNA methylation are highly context-specific, including lack of repression, as well as cases of increased gene expression, which appears to be driven by the eviction of methyl-sensitive transcriptional repressors. Furthermore, we find that some regulatory networks can override DNA methylation and that promoter methylation can cause alternative promoter usage. DNA methylation deposited at promoter and distal regulatory regions is rapidly erased after removal of the zinc finger-DNMT3A fusion protein, in a process combining passive and TET-mediated demethylation. Finally, we demonstrate that induced DNA methylation can exist simultaneously on promoter nucleosomes that possess the active histone modification H3K4me3, or DNA bound by the initiated form of RNA polymerase II.

**Conclusions:** These findings have important implications for epigenome engineering and demonstrate that the response of promoters to DNA methylation is more complex than previously appreciated.

**Keywords:** DNA methylation, Epigenome engineering, CpG islands, DNMT, Zinc finger, Promoter regulation

## Introduction

DNA methylation at the 5 position of cytosines has been associated with a plethora of biological roles in mammalian gene regulation, from cellular differentiation to genomic imprinting and X-chromosome inactivation [1, 2]. In most mammalian somatic cells, DNA methylation primarily occurs at CpG dinucleotides (mCG), where the majority of CpGs in the genome are fully methylated [3]. In contrast, regulatory regions generally remain unmethylated, most notably CpG-rich regions known as CpG islands (CGI) predominantly found at gene promoters [3–5]. DNA methylation at these CGI promoters is frequently inversely correlated with transcriptional activity [6–8]. Promoter hypermethylation is common in cancer and frequently associated with tumor-suppressor gene silencing [9, 10]. Mechanistically, DNA methylation is thought to interfere with transcription at promoters via preventing transcription factor binding or recruiting transcription repressor complexes [11–13]. Moreover, abolition of DNA methyltransferase activity through chemical inhibition or genetic disruption causes global demethylation and activates numerous genes [14, 15]. Such observations have led to the common conclusion that DNA methylation of CGI promoters and other regulatory sequences causes transcriptional silencing. However, these observations are largely correlative data. It is now becoming increasingly clear that the relationship between promoter methylation and gene transcription is more complex and context dependent than previously believed. DNA methylation has been reported to occur downstream of transcriptional regulation in various contexts. In early development and germline cells, transcription can occur from genes with methylated promoters [16–18], with recent reports from cancer cells supporting a similar conclusion [19]. In addition, transcriptional silencing can precede the acquisition of promoter DNA methylation [20], including genes on the female inactive X chromosome and imprinted genes during development [21, 22]. Similarly, across temporal series, transcriptional changes have been shown to usually occur prior to the demethylation of the regulatory region [23–25]. Furthermore, DNA methylation has been reported to recruit some transcription factors and paradoxically promote transcription [26–30]. Therefore, whether DNA methylation at promoters functions as a primary instructive biochemical signal for gene silencing remains unresolved.

Targeted methods to manipulate methylation state at endogenous loci have been developed in recent years [31–33], including customized zinc finger (ZF) domains fused to the catalytic domain of human DNA methyltransferase 3A (ZF-DNMT3A) [34–37] or the bacterial methyltransferase M.SssI [38], and more recently the fusion of DNMT3A and DNMT3B to transcription activator-like effector (TALE) proteins or nuclease inactive Cas9 [39–46]. These artificial epigenome modification techniques allow targeted interrogation of whether DNA methylation correlates with, or is causative for, transcriptional repression. Their application at a limited number of loci indicates that induction of promoter methylation is sometimes sufficient to repress transcription [47]. However, not all promoters show the same level of response to methylation induction. Widespread off-target methylation activity by ectopic or mutant DNA methyltransferases has been shown to be insufficient for broad repressive activity at many of the aberrantly methylated genes [48–50]. Therefore, approaches that allow targeted methylation at a genomic scale are crucial to understand the capacity of induced DNA methylation to cause changes in transcription at endogenous promoters in a systematic manner.

Given the efficient post-replicative maintenance of DNA methylation patterns by DNMT1 [51], DNA methylation has been implicated in long-term gene silencing [52]. However, the stability of targeted induced methylation appears to be variable, with examples of both stable [53] and transient [37, 44, 48, 54] methylation and silencing of different genes. Consistently, methylation levels of different types of genomic regions depend on the enzymatic balance between DNMTs and demethylating enzymes such as TETs [55–57]. This loss of methylation at regulatory elements questions our capacity to permanently silence genes using epigenome engineering tools relying on DNA methylation alone. Thus, a better understanding of what limits our capacity to deposit DNA methylation ectopically, what determines promoter response to DNA methylation, or the mechanistic links to methylation maintenance at regulatory elements is of paramount importance. To address these questions, we induced DNA methylation at thousands of regulatory regions throughout the human genome and performed a genome-scale investigation to determine the impact of this treatment on gene expression and chromatin state.

## Results

### ZF-D3A binding causes genome-wide DNA methylation gain

To design a tool to specifically deposit cytosine DNA methylation, we fused the DNMT3A catalytic domain to a previously generated artificial zinc finger (ZF-D3A-wt), originally designed to bind a GC-rich 18-bp sequence found at the *SOX2* promoter (Fig. 1A) [36]. As a negative control, we made an identical construct including four amino-acid mutations on the DNMT3A catalytic domain, known to abrogate the cytosine methyltransferase ability (ZF-D3A-mut, Additional file 1: Fig. S1A) [58]. Using a doxycycline inducible system, we stably introduced these constructs into MCF-7 cells to test the capacity of ZF-D3A-wt and ZF-D3A-mut to bind to and methylate the *SOX2* promoter. After 3 days of doxycycline induction, we collected cells expressing ZF-D3A by fluorescent activated sorting (FACS) purification of GFP-positive cells (Fig. 1B, Additional file 1: Fig. S1B), and we used ChIP-seq to confirm that both the ZF-D3A constructs bind to the *SOX2* promoter (Fig. 1C). Using whole genome bisulfite sequencing (WGBS) to profile DNA methylation, we observed a gain of 43% CpG methylation (mCG) across the *SOX2* promoter in the ZF-D3A-wt doxycycline-induced samples (Dox) (Fig. 1C), whereas no methylation gain was observed in the ZF-D3A-mut cells (Dox-mut). Promoter methylation in the Dox sample led to a 1.9-fold decrease in *SOX2* mRNA levels (FDR 5.9e−15), with no significant change in transcript abundance (FDR = 0.4) in the unmethylated Dox-mut sample, as assessed by RNA-seq (Fig. 1C). We then performed ATAC-seq to profile chromatin accessibility, which followed a similar pattern to the transcript abundance, showing accessibility loss exclusively in the methylated Dox samples (Fig. 1C). Overall, we validated our artificial zinc finger construct as an epigenome modifier capable of repressing *SOX2* via deposition of DNA methylation.

However, artificial zinc finger proteins are known to bind degenerate versions of their preferred target sequence, akin to endogenous transcription factors (TFs) [59, 60]. Screening our ZF-D3A-wt ChIP-seq data, we discovered that our zinc finger construct was enriched at 32,105 sites across the genome (MACS2 peaks, FDR < 0.05) (Fig. 1D). To test if such off-target binding could lead to a global induction of DNA methylation, we

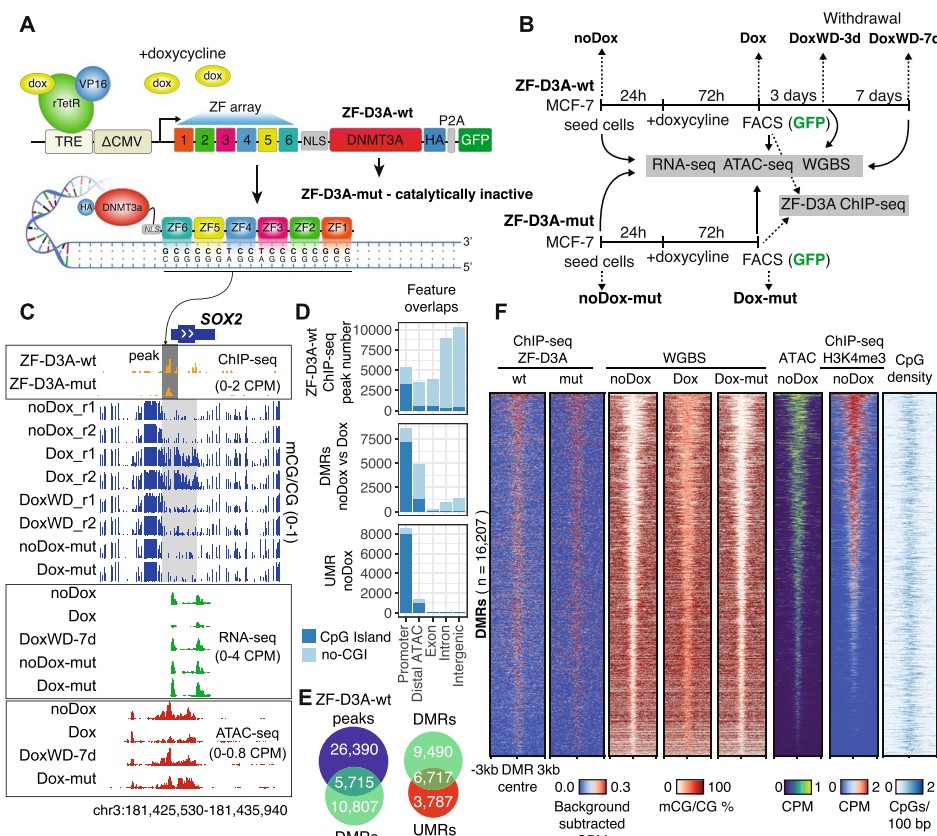

**Fig. 1** A synthetic zinc finger fused to D3A binds to thousands of loci leading to widespread 5mC gain. **A** Schematic representation of the ZF-D3A doxycycline inducible system. The ZF-D3A-mut has 4 amino-acid changes in the DNMT3A catalytic domain compared to the wild type (ZF-D3A-wt) that abrogate the DNA methyltransferase catalytic activity. The constructs only include the DNMT3A catalytic domain, not the full-length human gene. **B** Diagram of the experimental approach used in this study, highlighting stage names and sample harvesting points. After 3 days of Dox induction, only GFP expressing cells were collected for both Dox and Dox withdrawal (DoxWD) timepoints. For DoxWD-3d, no RNA-seq data was generated. **C** Genome browser representation of the *SOX2* locus. **D** Genomic distribution of identified ZF-D3A-wt binding sites, differentially methylated regions (DMRs) between ZF-D3A-wt noDox and Dox, and unmethylated regions (UMRs) in the ZF-D3A-wt control. Distal ATAC are ATAC-seq peaks > 2kb away from a TSS. **E** Overlaps between ZF-D3A-wt peaks and DMRs, and UMRs versus DMRs. Intersection values are shown for the top category (peaks and DMRs respectively). **F** Heatmap of the all DMRs and associated features, showing the binding signal of ZF-D3A constructs, mCG by WGBS, ATAC-seq and H3K4me3 ChIP-seq signal, and CpG density (*p*-value < 0.01, mCG gain >20%). Genome-wide consistency across epigenomic samples depicted in this figure is shown in Additional file 1: Fig. S13

characterized differentially methylated regions (DMRs) between the cells not expressing the zinc finger (noDox) and the Dox samples, identifying a total of 16,207 DMRs (dmrseq *q*-value < 0.05, > 20% mCG difference). Whereas ZF-D3A-wt peaks overlapped many types of genomic features, DMRs were predominantly found in gene promoters and distal regulatory elements, most frequently regions classified as CpG islands (Fig. 1D). The observed off-target methylation was not restricted to the zinc finger binding, since thousands of DMRs were found in regions without ZF-D3A-wt ChIP-seq peaks, indicating that methylation can be gained independently of the zinc finger (Fig. 1E), as commonly observed in dCas9-DNMT3-based approaches [39] Similarly, low-affinity zinc finger binding that is not reliably called as a peak, yet shows relative enrichment on the

heatmap, could indicate that some of these DMRs are bound by the ZF-D3A constructs (Fig. 1F). The majority (64%) of ZF-D3A-wt peaks that did not overlap with DMRs (Fig. 1E) were found in genomic regions with high (>80%) mCG in the noDox condition, which explains why no further methylation gain is detected with a DMR approach. A permutational test confirmed that the overlap between DMRs and ZF-D3A-wt was significant ($p = 0.0099$, regioneR), suggesting that ZF binding is strongly driving methylation gain.

To further understand how methylation is deposited upon ZF-D3A-wt induction, we used unmethylated regions (UMRs) in the noDox samples, which predominantly demarcate promoters encompassing CpG islands, to unambiguously identify loci of de novo methylation induction (Fig. 1D). The majority of UMRs overlapped with DMRs (Fig. 1E), indicating a widespread gain of methylation in these regions. However, the observation that 3787 UMRs did not overlap with DMRs suggests that the ZF-D3A-wt is not able to access the whole genome. Consistently, global methylation levels only show 1% increase in Dox samples compared to noDox (Additional file 2: Table S1). Thus, we observed gain of DNA methylation at thousands of sites, but not a global shift in methylation levels.

Overall, regions that gained methylation were enriched in ZF-D3A binding and open chromatin (Fig. 1F). To distinguish between active promoters and enhancers, we profiled the promoter-associated histone modification H3K4me3 using ChIP-seq. This showed that most, but not all, DMRs were in active promoters and that the DMRs that lack H3K4me3 binding also feature high CpG density. Therefore, the ZF-D3A-wt system allows the interrogation of induced DNA methylation at thousands of regulatory regions simultaneously, rather than laboriously testing one at a time.

Importantly, as in the *SOX2* example, we confirmed that ZF-D3A-mut recapitulates the binding patterns of ZF-D3A-wt but does not lead to widespread methylation induction (Fig. 1F). The ChIP-seq signal difference between ZF-D3A-mut and ZF-D3A-wt could correspond to differences in binding affinity, as previously shown when comparing an empty version of this zinc finger to a KRAB domain fusion version [59]. However, the difference in signal between ZF-D3A-mut and ZF-D3A-wt is likely driven by distinct signal to noise ratios in the ChIP experiments (Additional file 1: Fig. S2A). Consistently, when assessing the raw ChIP-seq mapping signal on peaks exclusively called in ZF-D3A-wt samples, we can also observe clear ZF-D3A-mut enrichment (Additional file 1: Fig. S2B). In contrast, ChIP-seq data of the empty ZF showed divergent binding activity. Since both ZF-D3A-mut and ZF-D3A-wt are on the top expressed transcripts after Dox induction (33rd and 66th highest respectively, Additional file 1: Fig. S1C), we do not favor lower transcription and decreased protein stability as a likely source of this difference in signal, but a combination of signal and slightly divergent binding preferences. Therefore, the mutant version of the methyltransferase provides a valid negative control for methylation activity while largely recapitulating binding capacities of the wild type construct.

### Nucleosomes restrict CG and CH methylation deposition by ZF-D3A

We then set out to understand how the ZF-D3A-wt binding properties and activity is affected by the native chromatin state. We first calculated the top enriched de novo motif in the ZF-D3A-wt ChIP-seq peaks (found in 75.9% of peaks, Fig. 2A), finding a

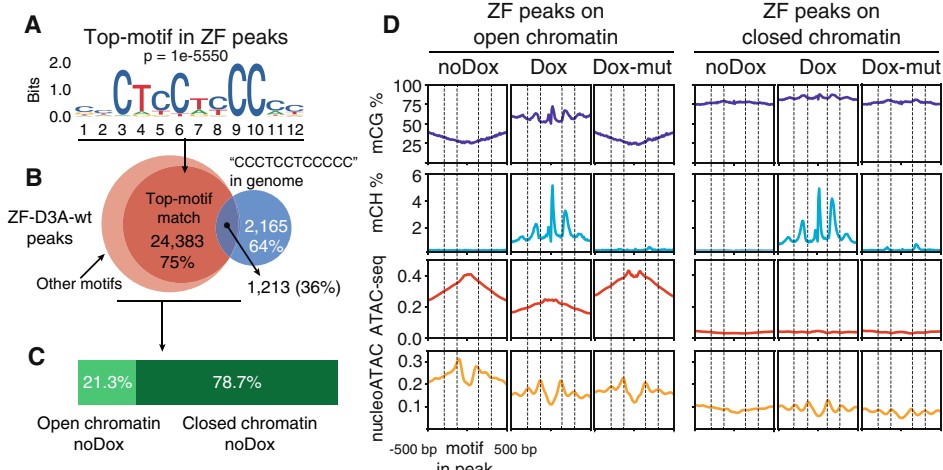

**Fig. 2** The ZF-D3A can bind closed chromatin yet 5mC deposition is limited by nucleosome positioning. **A** Top motif in ZF-D3A-wt ChIP-seq peaks. **B** Overlap between ZF-D3A-wt peaks, ZF-D3A-peaks harboring the top motif displayed in **A** (75.9% of all peaks), and genomic regions exactly matching the CCCTCCTCCCCC sequence in the human genome (hg19 build). **C** Percentage of ZF-D3A-wt peaks in open chromatin, defined as noDox ATAC peaks. **D** Mean distribution of CG methylation, CH methylation (H = A, T, C), raw ATAC-seq, and nucleoATAC signal on ZF-D3A peaks. ZF-D3A peaks are classified as per overlap with ATAC-seq peaks, and centered on the ZF top motif. Dashed lines indicate the periodic peaks corresponding to nucleosome positions defined by the nucleoATAC signal in the Dox state. noDox and Dox correspond to ZF-D3A-wt, whereas Dox-mut corresponds to ZF-D3A-mut

GC-rich motif with a consensus sequence that overlaps with 12 of the original 18 bp targeted by the zinc finger. This indicates that the zinc fingers 1 and 6 (Fig. 1A) are not essential for binding to the target site and that partial sequence targets are sufficient for ZF-D3A-wt binding.

Despite this broad binding capacity, only 36% of the genomic regions encoding an identical match to the 12 bp CCCTCCTCCCCC sequence were found within ZF-D3A-wt peaks (Fig. 2B). This suggested that some potential binding sites to ZF-D3A are not accessible to the construct, most likely because the native chromatin state was restricting, to some extent, the binding of this artificial ZF. To test this, we searched for ZF-D3A-wt peaks overlapping with regions exhibiting an open chromatin state (ATAC-seq peaks) in the noDox sample, since these represent the native accessibility landscape that the ZF-D3A encounters upon Dox induction. We found that the ZF-D3A constructs are capable of binding to both closed and open chromatin (Fig. 2C). In fact, the majority of ZF-D3A-wt peaks are in closed chromatin (78.7%), indicating that ZF-D3A could act as a pioneering factor.

To understand how distinct chromatin environments are affected by ZF-D3A-wt binding, we separately characterized open and closed chromatin regions. Accessibility decreased at the open chromatin regions bound by ZF-D3A-wt, coinciding with methylation gain (Fig. 2D). In contrast, regions bound by ZF-D3A-mut did not show a major accessibility loss, but rather just a footprint overlapping the ZF-D3A binding motif (Fig. 2D), indicating that the DNA methylation, and not zinc finger binding, is responsible for accessibility loss. Unlike in open chromatin, ZF-D3A binding does not lead to accessibility changes at closed chromatin regions, indicating that ZF binding on its own

is not enough to open chromatin on those sites, irrespective of the methylation capacity of the fused DNMT3 catalytic domain.

As TF binding is characterized by nucleosome displacement, we then assessed if nucleosome positioning was affected by ZF-D3A binding computed with nucleoATAC (see the "Methods" section, [61]). We found that nucleosomes neighboring the ZF-D3A-wt motif were displaced upon ZF-D3A binding, both in open and closed chromatin regions (Fig. 2D). This indicates that nucleosome displacement by ZF-D3A is not enough to make chromatin accessible, unlike endogenous pioneer TFs that recruit additional cofactors to open chromatin [62].

The capacity of DNMT3A and DNMT3B to methylate DNA is restricted by nucleosomes [63, 64]. Similarly, the methylation increase that we observed in open chromatin regions upon ZF-D3A-wt induction followed a pattern with inverse periodicity to the nucleosome positions (Fig. 2D). This suggests that nucleosomes also protect DNA from methyltransferases fused to epigenome modifiers, such as ZF-D3A-wt.

It has been previously suggested that epigenome modifiers using exclusively the DNMT3A catalytic domain are unable to deposit methylation in the CH context (where H = C, T, or A), allegedly due to the absence of the additional protein domains on the N-terminal of the native DNMT3A protein [48]. When measuring CH methylation surrounding the ZF-D3A binding sites, we observed appreciable deposition of CH methylation in both open and closed chromatin regions. Furthermore, CH methylation signal follows an inverse pattern to nucleosome positioning, mirroring the effects observed with CG methylation (Fig. 2D). Of note, CH methylation allowed higher resolution discrimination of nucleosome position than nucleosome signal from ATAC-seq data for the majority of ZF-D3A-wt binding sites. Signal from nucleoATAC and CH methylation validates the capacity of the artificial zinc finger to shift nucleosomes irrespective of the chromatin environment on a genomic scale (Additional file 1: Fig. S3A). Furthermore, the sequence preference for CH methylation recapitulates the TxCAC motif that characterizes native DNMT3A activity (Additional file 1: Fig. S3B) [65]. Overall, these data indicate that nucleosome displacement is achieved through ZF binding, yet the ZF-D3A-wt methyltransferase activity is restricted by nucleosomes. Therefore, the investigation of the transcriptional changes upon Dox induction requires careful discernment of the effects derived from ZF-D3A binding and those attributable to DNA methylation.

### Promoter methylation is not exclusively associated with gene repression

Since a majority of promoters gain methylation (Fig. 3A), we then wanted to test the transcriptional response upon ZF-D3A expression. As widespread gain in promoter DNA methylation could, in theory, silence most of the genes in the genome, this could reduce the total amount of mRNA per cell in the Dox samples [66]. Consequently, if the amount of RNA per cell was not the same across conditions, the assumptions of most statistical models designed to detect differential expression would be violated [67, 68]. To test if this type of global RNA reduction affected our data, we introduced a fixed amount of ERCC spike-ins per 50,000 cells, allowing us to test if the proportion of spike-ins versus cellular mRNA differed across conditions. If ZF-D3A-wt cells had less mRNA per cell due to widespread silencing, a higher proportion of ERCC spike-in relative to the cellular mRNA would be observed. However, the proportion of ERCC spike-in versus

mRNA did not show this trend; in fact, noDox and Dox showed indistinguishable values (Wilcoxon sum-rank test *p*-value > 0.05, Additional file 1: Fig. S4A, B). Furthermore, measuring the total RNA per sample (50,000 cells plus fixed ERCC spike-in), we did not find any evidence supporting a global shift of RNA per cell in the ZF-D3A-wt Dox samples (Additional file 1: Fig. S4C). Since the RNA-seq sequencing depth across genes and ERCC spike-ins was similar between replicates (average 48M counts/replicate, Additional file 1: Fig. S4D), this indicated we could use standard scaling normalization approaches to perform differential gene expression analysis.

We found 11,350 genes significantly differentially expressed between noDox and Dox conditions, whereas 5096 genes were significantly differentially expressed between noDox-mut and Dox-mut conditions. Since neither ZF-D3A-wt nor ZF-D3A-mut showed leaky expression prior to Dox induction (Additional file 1: Fig. S1C), we further tested our differential expression strategy comparing noDox and noDox-mut. As expected, we could only detect 13 differentially expressed genes in this comparison, of which only 5 had a fold change >2, confirming that our differential expression analysis is not inflated by noise and that both cell lines show equivalent expression levels in the noDox stage (Additional file 1: Fig. S4E). This indicates that doxycycline induction of both ZF-D3A-wt and ZF-D3A-mut results in a major transcriptional reconfiguration.

To discriminate the effects of methylation induction and ZF-D3A binding, we compared the differential expression changes between both constructs. The genes that were downregulated upon ZF-D3A-mut induction were predominantly downregulated upon ZF-D3A-wt induction, with a similar observation for upregulated genes (Fig. 3B). A direct comparison of fold change in expression between both constructs revealed a positive correlation (Spearman's *r* 0.68, Additional file 1: Fig. S4F), suggesting similar downstream transcriptional changes upon ZF-D3A binding. Next, we selected genes with promoters overlapping ZF-D3A-mut ChIP-seq peaks, to investigate if direct binding by ZF-D3A-mut lead to silencing. Fifty-five percent of these genes were downregulated upon expression of Dox-mut, but the remaining 45% were upregulated (Fig. 3C, Additional file 1: Fig. S5A). The secondary effects on non-directly bound genes were also mixed, showing both activation and silencing. This analysis enabled the ZF-D3A-mut transcriptional changes to be deducted from the ZF-D3A-wt comparisons to control for the ZF-D3A binding activity.

We then focused on genes that showed differential expression exclusively in the ZF-D3A-wt samples, since these transcriptional changes should be dependent on DNA methylation gain and not confounded by the effects of ZF-D3A binding. To test how promoter methylation affects gene transcription, we selected the most significant DMRs (*q*-value < 0.01) overlapping the promoters (promoter-DMRs) of genes that we can reliably detect in our RNA-seq data (>50 normalized counts in any given condition). We observed a trend toward transcriptional repression that was linked to methylation induction intensity (Fig. 3D, Additional file 1: Fig. S5B). However, a significant proportion of genes (46.5–48.6%) with methylated promoters did not show significant repression (FDR > 0.05) or were upregulated (Fig. 3E). Therefore, in this experimental system, promoter methylation is frequently insufficient for transcriptional silencing.

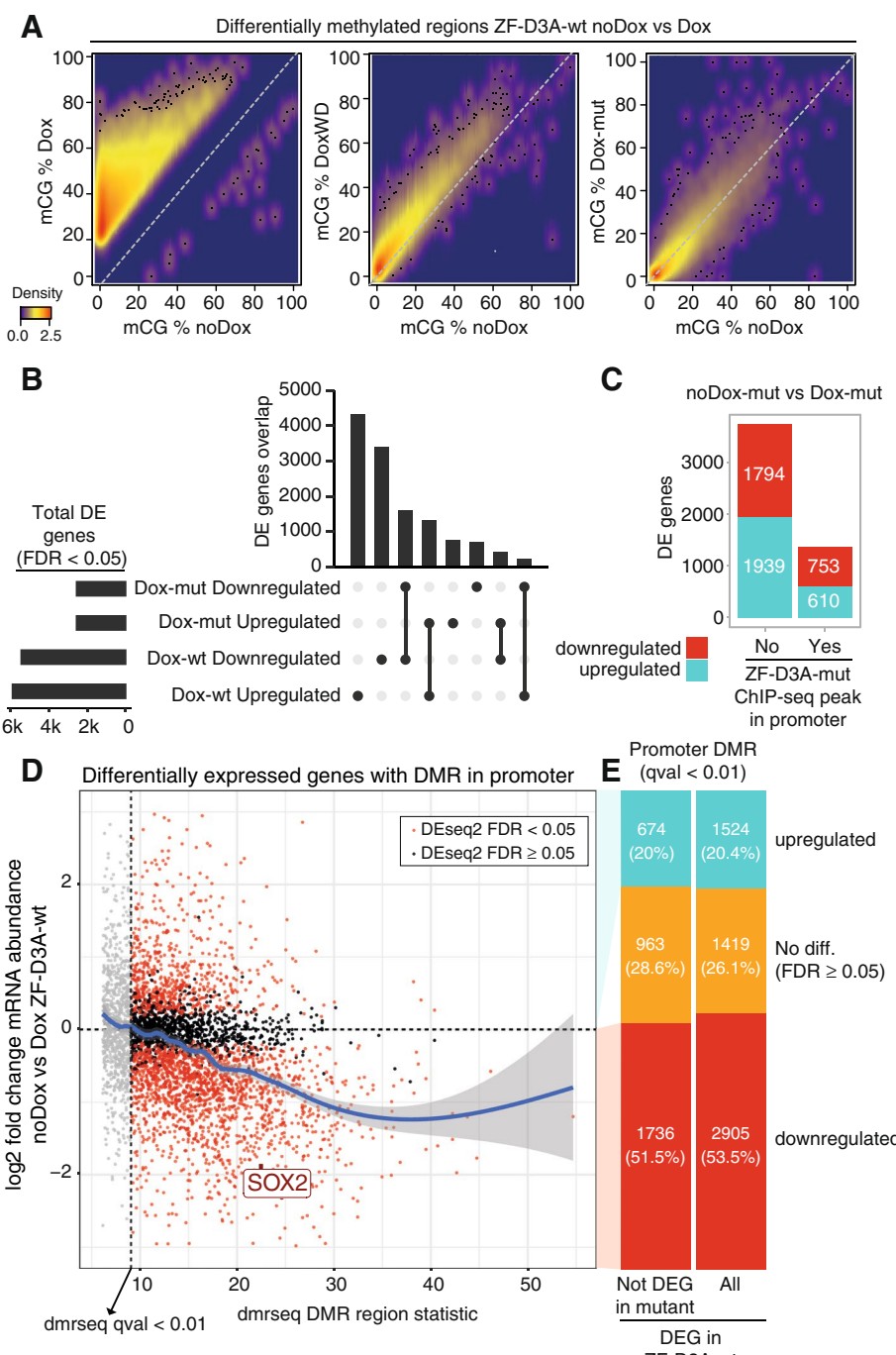

**Fig. 3** ZF binding and widespread methylation gain are not limited to transcriptional repression. **A** Heatmaps representing the mCG % in noDox versus Dox DMRs indicating the level of methylation gain in ZF-D3A-wt Dox, major loss of methylation on Dox withdrawal (DoxWD), and absence of methylation in ZF-D3A-mut expression (Dox-mut). **B** Overlap of differentially expressed (DE) genes (DEseq2 fdr < 0.05) between ZF-D3A-wt noDox vs Dox and ZF-D3A-mut noDox vs Dox. **C** Transcriptional change of differentially expressed genes classified as having a ZF-D3A-mut peak on the promoter (−2000, +500 bp). **D** Scatter plot of the DMR methylation gain (as defined by dmrseq statistic), versus the fold change in mRNA abundance of promoter-DMR associated genes, between noDox and Dox. Point color indicates the gene differential expression significance: red indicates FDR <0.05, black indicates FDR >0.05 (DEseq2). Trend line (blue) was fitted using a local polynomial regression. Depicted genes are not differentially expressed in the ZF-D3A-mut. **E** Proportion of genes with a promoter-DMR classified by transcriptional change. On the left, genes not differentially expressed in the ZF-D3A-mut and on the right all genes

**Transcription factor activity determines response to promoter hypermethylation**

Since transcriptional repression is the expected response of CGI promoter methylation [4], we wanted to understand if CGI presence in promoters underpinned the divergent transcriptional responses observed in promoter-DMRs. However, when inspecting all three types of gene expression change (downregulated, non-differentially expressed, and upregulated) in relation to promoter-DMRs, we observed that a majority of promoters (>87%) overlap with CGIs. This indicates that CGI presence alone does not explain the sensitivity of promoters to DNA methylation. We then tested if CpG density was different across promoter-DMRs, and discovered that downregulated genes feature higher median CpG density in promoters than upregulated or non-differentially expressed genes, albeit with largely overlapping distributions of CpG density (Fig. 4A). Similarly, DMR length and percentage methylation gain across DMRs were on average higher for downregulated genes (Fig. 4B, C). This suggests that higher and wider-spread methylation inductions contribute to the downregulation trend. Inspection of UMRs showed a similar pattern, where promoter-DMRs at downregulated genes tend to occupy a higher fraction of the UMR (Additional file 1: Fig. S6A, B). We then hypothesized that a lack of gene response to promoter methylation could be compensated by distal regulatory element activity. However, when inspecting the number of ATAC-seq peaks >2000 bp from TSS (indicative of regulatory elements), we found that downregulated genes featured the highest number of associated distal regulatory elements, followed by upregulated genes (Fig. 4D). Taken together, the highly overlapping distributions across all of these genomic features and levels of methylation induction indicate we should observe many examples of gene promoters with strong methylation induction without the corresponding transcriptional repression, which we indeed do (Additional file 1: Fig. S7). These observations exemplify the weakness of using general trends in promoter methylation change to explain all transcriptional responses.

As promoter-DMRs rarely achieve mean methylation gains above 40% (Fig. 4C), it is possible that genes that do not show repression upon Dox induction display higher cell-to-cell heterogeneity. In this scenario, non-repressed DMRs would have a higher proportion of cells not gaining methylation on the promoter compensated by stronger methylation in fewer cells, whereas downregulated genes would show more uniform methylation gain across cells. This scenario could lead to an apparent lack of transcriptional response as inferred from the bulk RNA-seq data. To test this, we calculated the proportion of linked methylated CpGs on DNA sequence reads overlapping promoter-DMRs (Fig. 4E). Once removing PCR duplicates (see the "Methods" section), sequenced read-pairs come from unique DNA molecules, representing values equivalent to single cells. When comparing the percentage of reads displaying full, intermediate, or lack of methylation at promoter-DMRs, we found that non-repressed genes showed the same percentage of unmethylated reads as the downregulated genes (Fig. 4E). Therefore, these data indicate that cell-to-cell heterogeneity in methylation is not the cause of the lack of repression we observe. Still, it is possible that positional methylation effects differentially affect each promoter-DMR type.

As chromatin accessibility at promoters influences gene transcription, we tested if accessibility differed across promoter-DMRs. We found that downregulated genes show the greatest loss of accessibility upon Dox induction, with a much lower

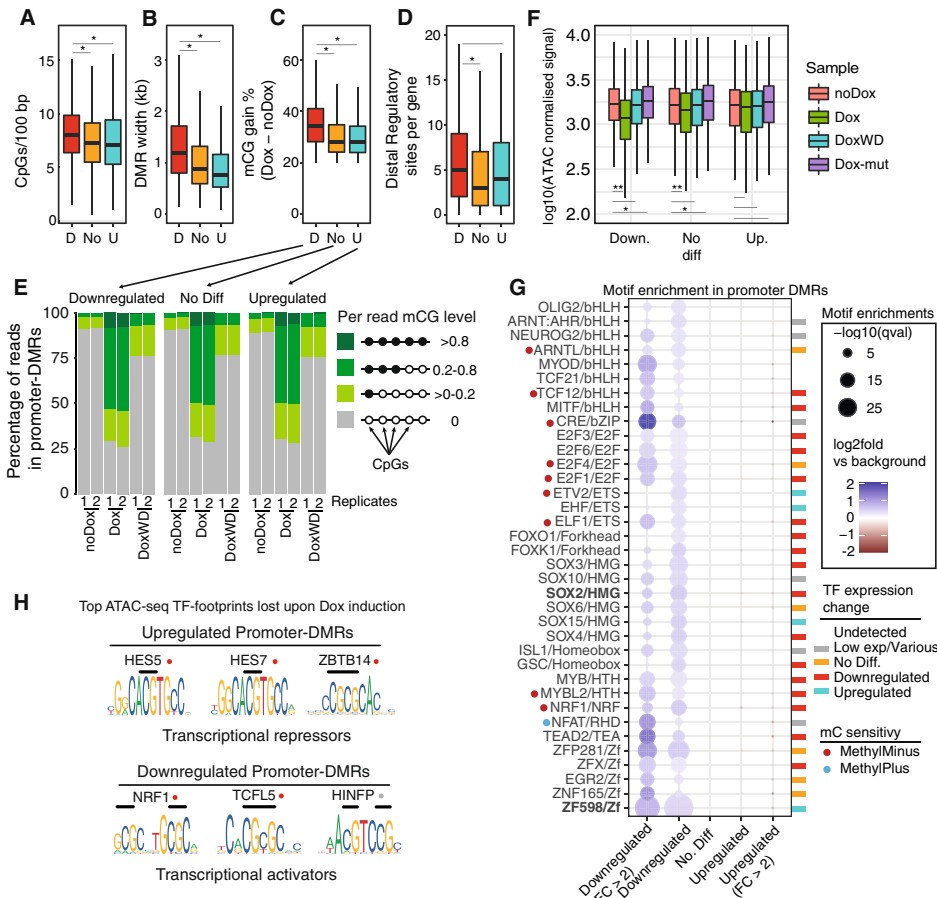

**Fig. 4** Transcriptional response to methylation gain is linked to differential transcription factor motifs. Distribution of **A** CpG density, **B** DMR width, **C** weighted average mCG % gain, and **D** number of distal regulatory sites per gene for promoter-DMR associated genes. Red depicts downregulated genes, orange depicts not differentially expressed genes (DEseq2 FDR > 0.05), and turquoise depicts upregulated genes in the noDox versus Dox comparison (Fig. 3E). In all four panels, an asterisk indicates a significant one-sided Wilcoxon rank-sum test ($p < 0.01$), and lack of asterisk indicates a higher $p$-value. **E** Proportion of reads at promoter-DMRs classified by per-read mCG %. Only reads spanning at least 5 CpGs were used. **F** ATAC-seq signal on DMRs across different samples classified by transcriptional response. ATAC-seq signal normalized using DEseq2. One asterisk indicates a $p$-value < 0.01 in a two-sided Wilcoxon rank-sum test between two distributions, and two asterisks indicates a $p$-value < 0.001. Boxplot center lines are medians, box limits are quartiles 1 (Q1) and 3 (Q3), whiskers are 1.5 × interquartile range (IQR), and points are outliers. **G** Transcription factor binding motif enrichments in promoter-DMRs classified by transcriptional change. Bonferroni-corrected $p$-values obtained from HOMER2 represented by circle size and color depicts log fold level of enrichment versus background. Colored rectangles indicate the transcriptional status of the associated transcription factor, where gray indicates that there are many TFs associated with the motif or low expression level (<50 baseMean DEseq2 normalized level). Red and blue dots indicate TFs shown to be repelled or attracted by 5mC respectively. Highlighted in bold are SOX2 (original target of the ZF) and the ZF-D3A motif. **H** Highest scoring motifs associated with the TF footprints depleted upon Dox activation at promoter-DMRs, derived from ATAC-seq data. CpGs on the core-binding motifs are highlighted with black lines, and red circles indicate those previously reported to be repelled by methylation

magnitude for loss for non-repressed genes, and negligible changes for the ZF-D3A-mut control (Fig. 4F, Additional file 1: Fig. S6C, F). The difference in promoter accessibility did not result in an obvious change in nucleosome patterning around the TSS at downregulated genes (Additional file 1: Fig. S6C). It was also possible that silencing

is restricted to promoters accumulating some type of repressive histone modification, such as H3K9me3 or H3K27me3. We performed ChIP-seq of these repressive marks and observed that H3K9me3 is rare and not induced upon methylation induction, yet H3K27me3 is deposited upon Dox induction, consistent with other ZF-D3A reports (Additional file 1: Fig. S6D) [37]. Intriguingly, genes with promoter-DMR that tend to get upregulated show higher levels of H3K27me3 than the rest (Additional file 1: Fig. S6D). Therefore, transcriptional change does not appear to be related to nucleosome repositioning, but to an overall loss of accessibility at promoters and likely linked to H3K27me3 deposition.

We then asked if the underlying regulatory networks could be a major determinant of the transcriptional response to promoter-DMRs. Since many TFs are methylation sensitive [12, 28, 29], we tested for TF binding motif enrichment for the different promoter-DMRs. We found that downregulated genes were enriched for numerous TF motifs compared to the rest of the promoter-DMR gene classes, which showed no significant enrichments (Fig. 4G, Additional file 1: Fig. S6E). Among these motifs, we observed known methyl-sensitive TFs such as NRF1, bHLH, or ETS [13, 28, 69], indicating that these TFs might be repelled by the DNA methylation deposited upon Dox induction. Other motifs were linked to TFs that are downregulated upon Dox induction, such as *SOX2*, but the methyl-sensitivity for most of those motifs is not yet known or they lack CpGs directly within their core-binding motifs. This suggests that many of the transcriptional changes observed for promoter-DMR genes could be *trans* secondary effects unlinked to proximal methylation gain. Finally, promoter-DMRs of downregulated genes were enriched for the ZF-D3A-wt binding motif, suggesting multiple binding events of the zinc finger may contribute to transcriptional repression (Fig. 4G). These differences suggest that repressed promoters harbor, on average, more regulatory information in common, than those less responsive to methylation gain.

We then assessed if TF binding could be affected by methylation induction through leveraging the ATAC-seq data to find TF footprints on these promoter-DMRs, finding that promoter-DMRs show evidence of pronounced TF footprint loss in the Dox state (Additional file 1: Fig. S8). However, the TFs associated with these reduced footprints are not the same across the 3 types of promoter-DMRs. Downregulated gene promoter-DMRs display NRF1 as the most depleted TF, followed by ZBTB14, HINFP, and TCFL5 (Fig. 4h). In contrast, the most depleted footprints for the upregulated gene promoter-DMRs are assigned to HES5/HES7 and ZBTB14, with a differential binding score above > 0.4, much more pronounced than any other footprint (Fig. 4H, Additional file 1: Fig. S8). All these TF motifs display several CpGs on their core, and all but HINFP have been experimentally shown to be repelled by methylation [13, 28]. Importantly, HES and ZBTB14 are transcriptional repressors [70, 71], which suggests that the observed upregulation of the genes driven by these promoters could be attributed to these repressors being repelled by induced methylation, thus facilitating transcription. This is consistent with most upregulated genes exhibiting low transcript abundance in the noDox state and upregulated promoters not showing accessibility loss in Dox (Fig. 4F, G, H). Overall, these findings indicate that specific TF classes play a major role in the transcriptional response to induced promoter methylation, both through repelling transcriptional

activators leading to silencing, and the unexpected activation of gene expression by repulsion of repressors. This reinforces the view that the effects of DNA methylation induction are not always repressive, emphasizing that the transcriptional responses to induced promoter methylation are highly context dependent.

### Promoter methylation leads to alternative promoter usage

DNA methylation has previously been reported to play a role in alternative promoter usage [72, 73]. Therefore, we hypothesized that in this system, forcibly methylating promoters could be compensated for by opening alternative promoters. Consistent with this, we found 261 genes showing differential promoter usage between noDox and Dox RNA-seq samples, which we defined as the "primary TSS" for the basal TSS and the "secondary TSS" for the newly active TSS (Fig. 5A). The promoters associated with the primary TSS mostly overlapped with CGIs (87%), while a minority of secondary TSS did (39%) (Additional file 1: Fig. S9A). Since ZF-D3A-wt binds CpG-rich regions such as CGIs, we found that primary TSS had higher methylation induction than secondary TSS upon ZF-D3A-wt expression (Fig. 5B), likely driving secondary promoter activation. Consistently, primary TSS exhibited lower accessibility upon DNA methylation induction, whereas secondary TSS gained accessibility (Fig. 5C). Notably, secondary TSS had sparse H3K4me3 signal at the noDox stage and showed higher levels of H3K4me1 in MCF-7 cells than the regions surrounding the primary TSS, suggesting that secondary TSS lacked the typical epigenomic characteristics of an active promoter prior to Dox

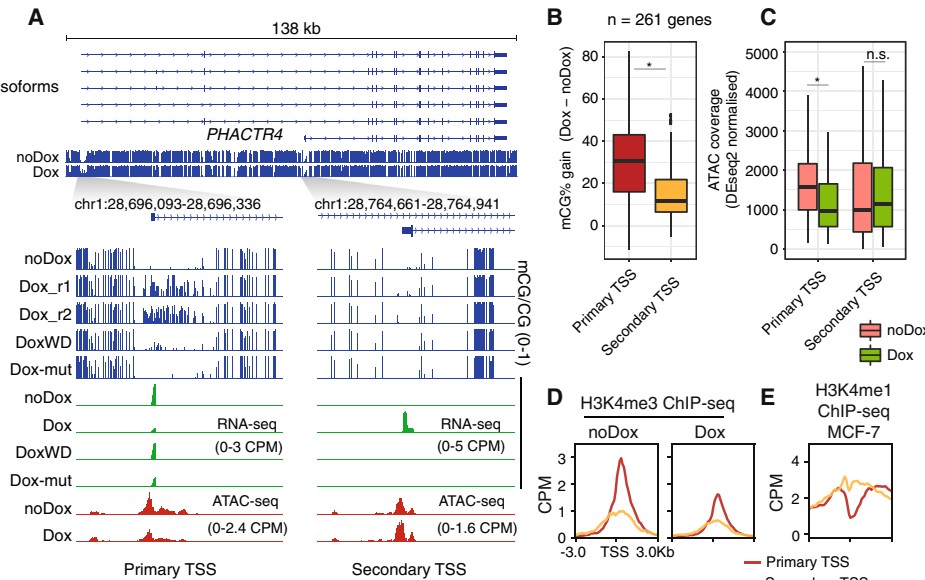

**Fig. 5** Methylation induction leads to differential promoter usage. **A** Genome browser snapshot depicting *PHACTR4* known isoforms, focusing on the primary transcriptional start site (TSS) in noDox cells and the secondary TSS site that appears upon Dox induction. Distribution of **B** mCG% gain and **C** ATAC-seq signal on primary and secondary TSS sites (spanning ±500 bp from TSS). One asterisk indicates a *p*-value < 0.01 in a two-sided Wilcoxon rank-sum test between two distributions, while n.s. depicts a higher value. Boxplot centre lines are medians, box limits are quartiles 1 (Q1) and 3 (Q3), whiskers are 1.5 × interquartile range (IQR), and points are outliers. Average **D** H3K4me3 and **E** H3K4me1 signal on the primary and secondary TSS. H3K4me1 MCF-7 ChIP-seq was obtained from the ENCODE database

induction (Fig. 5D, E), but rather more closely resembled the features of an enhancer. Importantly, these regions are depleted of H3K27me3, and thus not suggestive of a poised state (Additional file 1: Fig. S8A). Yet, both primary and secondary promoters display accessibility in both stages, indicating that alternative promoter usage is achieved at accessible regulatory elements (Additional file 1: Fig. S9B). As for promoter-DMRs, differential TF motifs and higher ZF-D3A-wt binding discriminate between the primary and secondary TSS, suggesting that a combination of zinc finger binding and different TF repertoires dictate promoter usage (Additional file 1: Fig. S9C).

In contrast, the ZF-D3A-mut only showed 21 genes with differential TSS usage, of which only 10 overlapped with those affected by ZF-D3A-wt expression (Additional file 1: Fig. S9D). This suggests that most of the TSS switching is dependent on DNA methylation induction and not simply due to the ZF expression.

The gene-level transcriptional response to the alternative promoter usage was not always enough to maintain the base levels of transcription before Dox induction. We found 130 genes that were downregulated, 47 were not differentially expressed, and 74 upregulated. Together, these results indicate that cells can, in some cases, compensate for forced methylation of CGI promoters by repurposing intragenic enhancers as alternative promoters.

### Methylation induction is compatible with transcriptionally active chromatin

It is reasonable to expect widespread methylation gain will lead to widespread reduction in chromatin accessibility. However, as we observed that many genes are upregulated upon Dox induction, we hypothesized that at least some of that response should come from activation of alternative regulatory networks. When gathering all the ATAC-seq peaks across all samples and performing differential peak calling, we only identified one differential peak when comparing noDox to Dox-mut (Fig. 6A, Additional file 1: Fig. S10A). This suggests that chromatin accessibility remains unaltered without the intervention of induced DNA methylation, despite thousands of genes being differentially expressed upon ZF-D3A-mut expression (Fig. 3B). Similarly, DoxWD showed no differential peaks, suggesting a reversion to the noDox state in terms of accessibility (Fig. 6A). In contrast, in the noDox versus Dox-wt comparison, we observed 18,841 differential peaks (FDR < 0.05, Fig. 6A). The vast majority of peaks that intersect with DMRs lose accessibility in the Dox-wt condition (5465 vs 596), confirming the role of methylation in closing chromatin (Fig. 6A). However, 9367 peaks gained accessibility upon Dox induction, indicating that new regulatory sites opened despite the widespread methylation induction (Fig. 6A).

To understand the differences between regulatory sites that gain or lose accessibility upon ZF-D3A-wt induction, we then analyzed methylation and accessibility changes in these regions. On average, sites that lose accessibility had lower methylation and had higher accessibility prior to Dox induction (Fig. 6B). In contrast, sites that gain accessibility upon Dox induction had lower accessibility in the noDox stage and did not show significant methylation gain, with mean methylation of ~50% in both noDox and Dox conditions (Fig. 6B). This indicates that many methylated sites gain accessibility upon Dox induction without being directly affected by the methylation status of the DNA (Additional file 1: Fig. S10B, C). This recapitulates previous findings where DNA

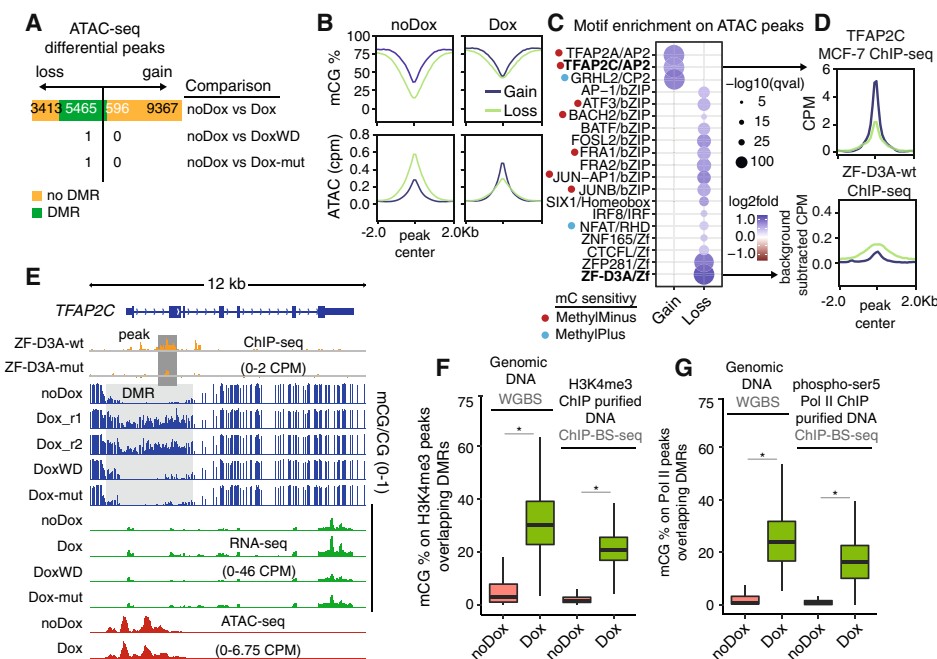

**Fig. 6** Active chromatin is compatible with DNA methylation. **A** Number of differentially called ATAC-seq peaks (DEseq2 FDR < 0.05) classified as gain or loss with respect to the noDox sample, DoxWD corresponds to 7-day withdrawal. **B** Average methylation and ATAC signal and **C** TF binding motifs at differential ATAC-seq peaks in noDox versus Dox conditions. **D** TFAP2C (MCF-7 ENCODE) and ZF-D3A-wt ChIP-seq signal on differential ATAC-seq peaks, color coded as in **B**. **E** Genome browser snapshot showing the *TFAP2C* locus, where DoxWD corresponds to 7-day withdrawal. Average methylation levels on **F** H3K4me3 and **G** RNA polymerase II phosphorylated serine 5 peaks called from bisulfite-ChIP-seq data. On the left hand side are the methylation values obtained as per WGBS and on the right hand side as per BS-ChIP-seq. One asterisk indicates a *p*-value < 0.01 in a one-sided Wilcoxon rank-sum test. Boxplot center lines are medians, box limits are quartiles 1 (Q1) and 3 (Q3), whiskers are 1.5 × interquartile range (IQR), and points are outliers

methylation co-exists with open chromatin during early stages of transcriptional change [23–25].

We then used motif enrichment analysis to predict the TFs responsible for the Dox-induced newly accessible sites and to predict TFs repelled by DNA methylation on sites that lose accessibility. We observed that accessibility gain peaks were enriched in AP2 and GRHL2 TF binding motifs (Fig. 6C). Despite TFAP2A and TFAP2C having been shown to be methyl-sensitive in vitro [28], there are no CpG sites in their core-binding motif, which indicates that in vitro methyl-sensitivity is not sufficient to predict binding on native chromatin. The peaks that lose accessibility were enriched in bZIP TFs, SIX, or IRF, and most prominently by the ZF-D3A motif (Fig. 6C). This was consistent with a higher ZF-D3A-wt binding in these regions (Fig. 6D). Similarly, we confirmed stronger TFAP2C binding enrichment at Dox-induced newly accessible sites using available ChIP-seq data for MCF-7 cells (Fig. 6D). Similarly, differential TF footprints between Dox and noDox largely agree with the motif enrichment data, showing TFAP2C and GRHL2 footprints enriched in the Dox condition, and JUN/FOS bZIP footprints in the noDox condition (Additional file 1: Fig. S10D, E). Interestingly, *TFAP2C* had the strongest promoter-DMR for any TF (*q*-value = 0.00027), yet

showed a 1.2-fold increase in transcription upon Dox induction (Fig. 6E), confirming that some regulatory pathways are not repressed by DNA methylation.

DNA methylation and H3K4me3 are typically mutually exclusive marks in CpG-rich regions [5]. However, it is not known whether forced DNA methylation deposition is sufficient to induce the loss of H3K4me3. We found that H3K4me3, on average, decreased upon methylation induction at promoters overlapping DMRs (Fig. 4H), recapitulating observations of a similar study using another zinc finger fused to DNMT3A [37]. However, because of the limited sensitivity of ChIP-seq and the possibility that there could be a heterogeneous population of loci with methylated and unmethylated DNA molecules, we directly measured the DNA methylation level in H3K4me3-modified chromatin using ChIP-bisulfite-sequencing (ChIP-BS-seq) [74, 75]. We identified H3K4me3 peaks and measured methylation levels using both WGBS data and the ChIP-BS-seq (Fig. 6F). In DNA purified by H3K4me3 ChIP in ZF-D3A-wt Dox, a median increase of 20.8% of methylation was observed, showing similar levels as for Dox WGBS, demonstrating that H3K4me3 and DNA methylation can exist simultaneously at the same site upon forced induction of DNA methylation.

As promoter DNA methylation could interfere with RNA polymerase II activation, we investigated whether the initiated form of RNA polymerase II (phosphorylated at Serine 5 of the C-terminal Domain; phospho-Ser5) was able to bind DNA methylated by ZF-D3A-wt, by ChIP-BS-seq of genomic DNA isolated with an anti-phospho-Ser5 RNA polymerase II antibody. We analyzed the methylation status of phospho-Ser5 RNA pol II ChIP-seq peaks (Fig. 6G), and similar to H3K4me3, there was only a small decrease in the median DNA methylation level between non-immunoprecipitated bulk genomic DNA upon Dox induction (median mCG 24%) and DNA bound by phospho-Ser5 RNA pol II (16%). Importantly, phospho-Ser5 RNA pol II was clearly able to directly interact with ZF-D3A methylated DNA. Together, these data demonstrate that forced DNA methylation is not sufficient to disrupt H3K4me3 occupancy or the interaction of initiated RNA polymerase II with genomic DNA.

### Methylation retention is linked to transcription factor binding and chromatin accessibility

The methylation status of genomic regions depends on the equilibrium between de novo methylation activity by DNMT3 enzymes, the maintenance of methylation by DNMT1, and demethylation, which can be driven either by TET oxidation and passive mechanisms coupled to cell division [55, 56]. We observed that most DMRs lose methylation after 3 days of Dox withdrawal, indicating that demethylation prevails at those regulatory sites (Fig. 3A). However, full demethylation reverting to the noDox state was not achieved at all DMRs after 7 days of withdrawal (Fig. 7B). To understand what dictates these demethylation differences, we classified DMRs into two categories: DMRs that lose methylation (loss-DMR, see the "Methods" section) and DMRs that retain methylation (retain-DMR) (Fig. 7B). Loss-DMRs are more common than retain-DMRs in number, and most loss-DMRs are found in gene promoters (65.8%, Fig. 7C, Additional file 1: Fig. S11A, B). In contrast, a higher percentage of retain-DMRs are found in distal regulatory elements (41.2% in retain-DMRs vs 34.2% in loss-DMRs). This revealed that methylation loss is the norm across forcibly methylated regions, yet demethylation pace is not the same across regions.

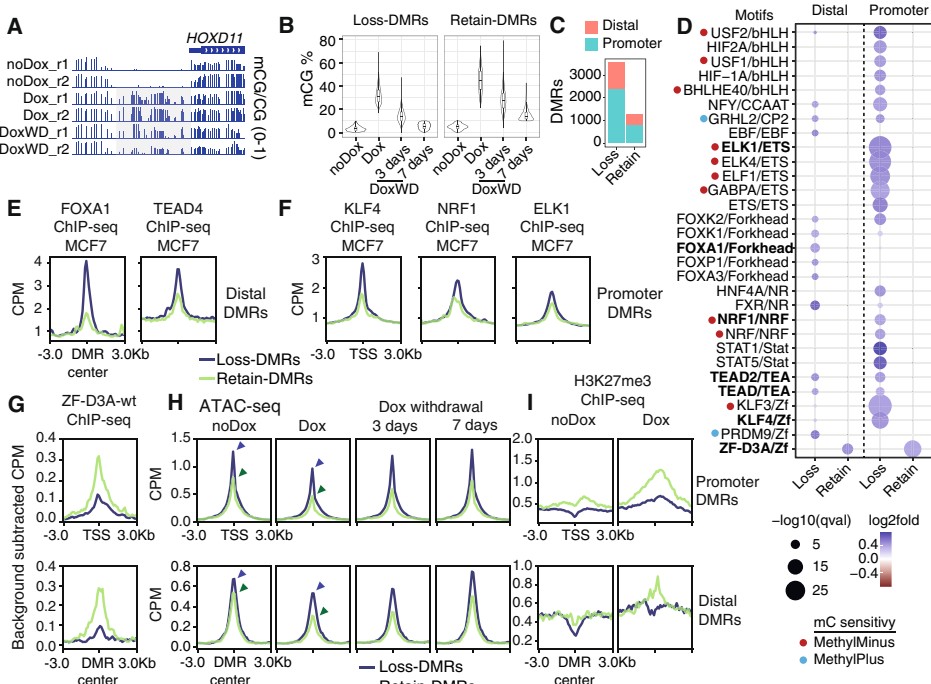

**Fig. 7** Rapid methylation loss is driven by accessibility and transcription factor binding. **A** Genome browser snapshot depicting a recalcitrant DMR on the *HOXD11* promoter. **B** Methylation levels in DMRs with a trend of 5mC loss (mCG on DoxWD-7days < 10%) or 5mC retain (mCG on DoxWD-7days > 10%). **C** Number and overlap of DMRs classified per methylation retention trend. **D** TF binding motif enrichments in DMRs, highlighted in bold, are those motifs confirmed by differential ChIP-seq signal. Average ENCODE MCF-7 TF ChIP-seq signal on **E** distal regulatory elements and **F** promoters. **G** ZF-D3A-wt ChIP-seq signal (background subtracted), **H** ATAC-seq signal, and **I** H3K27me3 levels (CPM) upon Dox induction at promoters and distal regulatory elements classified as per methylation retention trend. Dark blue lines depict DMRs that lose methylation and green lines depict DMRs that retain methylation

When inspecting enriched sequence motifs at loss-DMRs, we observed multiple TF motifs that distinguish promoters and distal regulatory elements (Fig. 7D). Promoters were enriched for motifs such as NRF or ETS, whereas distal regulatory elements were characterized by Forkhead or PRDM9 motifs (Fig. 7D). To validate these predictions, we used publicly available MCF-7 ChIP-seq datasets and found that distal loss-DMRs had stronger binding of FOXA1 and TEAD4 than DMRs that retain methylation (Fig. 7E). Similarly, loss-DMRs at promoters showed higher binding of NRF1, KLF4, or ELK1 compared to retain-DMRs (Fig. 7F). This demonstrates that TF binding activity is associated with faster demethylation at regulatory regions.

In contrast, DMRs that retain methylation did not show any specific TF motif enrichment besides the ZF-D3A motif (Fig. 7D). Consistently ZF-D3A-wt binding was higher at retain-DMRs (Fig. 7G). Retain-DMRs also tended to accumulate stronger methylation upon Dox induction (Fig. 7B). Furthermore, retain-DMRs tend to accumulate more H3K27me3 than loss-DMRs (Fig. 7I). Therefore, this indicates that stronger ZF-D3A binding coupled with higher methylation and H3K27me3 gain results in slower methylation removal.

Beyond the binding of ZF-D3A-wt, we wanted to understand if accessibility could influence the rate of methylation loss. In the noDox state, loss-DMRs displayed

higher accessibility (Fig. 7H). In contrast, retain-DMRs had lower accessibility, which showed stronger depletion upon Dox induction. Distal regulatory elements showed a similar pattern, yet the initial difference at the noDox state was less pronounced than promoters, and the relative accessibility loss was more pronounced in loss-DMRs. In most cases, accessibility returned to noDox levels after 7 days of withdrawal (Fig. 7H). Consistently, expression levels of genes that have promoter-DMRs that lose methylation were higher than those that retain it (Additional file 1: Fig. S11C). For instance, *HOXD11* is not expressed in noDox (0 read counts, Fig. 7A). Therefore, overall, stronger transcription and higher accessibility facilitate the removal of methylation at these sites.

To understand the relative contribution of TET-dependent demethylation across distinct DMR types, we performed Tet-assisted bisulfite sequencing (TAB-seq) [76] to map hydroxymethylated cytosines (hmC) upon Dox induction. Despite TET2 and TET3 being downregulated upon Dox induction (Additional file 1: Fig. S11D), we observed a peak of hydroxymethylation at regulatory sites (Fig. 7I), which was indistinguishable between loss-DMRs and retain-DMRs (Additional file 1: Fig. S11E, F). In contrast, promoters and distal regulatory elements that do not overlap with DMRs show significantly lower hmC levels than both retain-DMRs and loss-DMRs ($p$-value $< 1e-13$, one-sided $t$-test, Additional file 1: Fig. S11F). This indicates that TET enzymes are actively oxidizing the newly methylated CpGs at promoters and distal regulatory elements, yet do not fully predict the methylation maintenance. Overall, this data suggests that the rate of methylation loss at regulatory elements is dependent on several factors, but TF binding is likely to play a role in both TET-dependent and passive demethylation.

## Discussion

DNA methylation is considered to be a stable and repressive modification, and targeted approaches to manipulate methylation states are being explored for both research and clinical applications [43, 44]. However, the potential for DNA methylation to function as a primary instructive signal for transcriptional silencing of nearby genes remains largely unknown. Here, we show that the simultaneous methylation of thousands of promoters in the human genome frequently resulted in no detectable repression of gene expression. Although DNA methylation mostly decreased chromatin accessibility across regulatory regions, in most cases, it was not sufficient to reconfigure DNA into a stable heterochromatinized state, and at only a subset of regions do we see some accumulation of H3K27me3. We found that active genomic marks, such as initiated RNA pol II and H3K4me3, are able to co-exist with DNA methylation, implying that these chromatin states are not mutually exclusive with DNA methylation. Despite observing a frequent (~50%) association between promoter DNA methylation and repression, many genes with CGI in their promoters escape this trend, strongly suggesting that regulation by DNA methylation is more complex than previously appreciated. We find evidence that genes that become upregulated upon methylation induction are enriched for methyl-sensitive transcriptional repressors, which could partially explain why these genes gain expression instead of losing it. This indicates that the observed context-specific roles of DNA methylation are heavily dependent on TF binding affinities, where repulsion of TF activators such as NRF1 can lead to repression, but it can also lead to activation,

which has not been broadly reported to date. Our observation that context-specific epigenomic features influence the effect and retention of DNA methylation is extrapolable to CRISPR-based technologies, as recently reported in K562 cells [77].

Arguably, systems that methylate thousands of regions at once, such as the ZF-D3A-wt system, display inherent limitations that challenge the universality of our observations. Methylation gain in promoters could require binding of methyl-CpG binding proteins to attract repressive complexes to induce long-term silencing. Under such a model, widespread repression by artificial methylation induction could potentially not be achieved effectively due to lack of available proteins in the cell. However, our observation that some genes are more likely to be repressed than others is crucial to better understand what determines promoter methylation sensitivity, and provides a uniquely tractable system to study TF methyl-sensitivity in native chromatin [39]. Another challenge for these systems derives from secondary effects linked to silencing (or activation) of TFs affecting transcriptional activity of multiple downstream genes. Many of these effects are unlikely to affect CRISPR-based approaches aimed at one or few loci. Similarly, ZF-D3A binding can compete for binding sites with endogenous TFs, an effect that we control for using the catalytically inactive ZF-D3A-mut version. The competition between the ZF-D3A or ZF-D3A-mut and TFs might be entangled with the methyltransferase activity, since methyl-sensitive TFs would be outcompeted only by catalytically active ZF-D3A. However, here, we restrict our observations to promoters with strong methylation induction, and we still frequently observe genes that are not repressed, even finding genes that are upregulated. DNA methylation is unlikely to be acting as an activator in those later cases, but it rather demonstrates that some regulatory networks can override DNA methylation, which is in line with multiple observations in which methylation is lost or gained after transcriptional change occurs [23–25]. In CRISPR-based approaches, the binding competition with endogenous TFs might differ from that of artificial ZFs. Another challenge to the universality of this system derives from the MCF-7 cell line, which is a highly derived cancerous cell line. However, previous reports from genome-wide methylation inductions suggest similar tendencies using other systems or cell lines, such as HEK293, 293T, or mouse ESCs [37, 48, 49]. Finally, we do not achieve full methylation of most loci, which is an acknowledged limitation to most epigenome engineering approaches [31]. But this limitation likely indicates that demethylation of active regulatory sites is too strong for achieving forced silencing through methylation alone.

Our observations have multiple critical implications for epigenome engineering approaches. First, we provide a list of promoters that are more likely to be repressed by DNA methylation (Additional file 2: Table S2). These promoters have a set of characteristic sequence features, including a combination of overrepresented TF binding motifs, high CpG densities, or wider UMR, which could be used to design more effective epigenome editing-based gene targeting strategies. However, when targeting non-promoter regions or CpG poor promoters, some of these features might not be highly predictive of silencing. Binding of methyl-sensitive repressors should also be taken into account to avoid undesired activation effects. Furthermore, we demonstrate that nucleosomes physically impede methylation deposition on native chromatin when using DNA-binding-DNMT fusion proteins such as ZF-D3A-wt. Even in downregulated promoter-DMRs, we do not observe a major reconfiguration of nucleosome positioning around

the TSS, which clearly represents a roadblock to methylation induction strategies relying on fusion DNMTs with either CRISPR or ZF effectors. These challenges are more likely to be bypassed through epigenome editing approaches relying on in vitro methylation coupled to homologous recombination [78, 79], or simultaneous recruitment of other factors that alter nucleosome positioning. Additionally, we show that methylation of the primary promoter of many genes leads to activation of intragenic secondary promoters, which needs to be accounted for when designing promoter-silencing approaches.

The widespread loss of induced methylation at regulatory regions challenges the stability of CpG methylation as a silencing mark. Our observation that most DMRs decrease their methylation levels after 7 days of withdrawal indicates that strong demethylation activity is found both at promoters and at distal regulatory regions. Methylation loss is already noticeable at day 3 post-Dox withdrawal, yet not as pronounced, which stresses the importance of sampling long-term timepoints to assess methylation stability. Lack of methylation retention has been recently reported using similar strategies of genome-wide methylation induction in 293T and HEK293 cells [37, 48] or in targeted CRISPR-based approaches (reviewed in [47]). Our data suggest that hydroxymethylation and TET-mediated demethylation activity is playing a role in keeping regulatory regions demethylated, yet it might be insufficient to fully explain the rates of methylation loss. Regions that tend to lose methylation faster show similar hydroxymethylation levels to those that retain some methylation. Finding that TF binding and chromatin accessibility is higher in the regions that lose methylation faster lends support to models linking TF binding to accelerated passive demethylation. Complementary observations from a ZF-DNMT3A approach in HEK293 cells suggests that H3K4me3 levels and sequence-specific features protect CGIs from methylation, whereas H3K27me3 contributes to methylation stability [37]. Recently, DNMT1 has been shown to lack maintenance fidelity at regulatory elements [55, 57], and active TF binding has been proposed as a mechanism impeding DNMT1 fidelity [55]. Together, these data shed new light on methylation equilibrium dynamics at regulatory elements and also highlight the major role that TFs have in demethylation. Despite some TFs being methyl-sensitive and repelled by methylation, many are insensitive [13, 28], which would allow these TFs to bind the forcibly methylated regions and demethylate them again. This has profound implications for strategies relying only on transient DNA methylation deposition as a long-term silencing mechanism. Recent reports suggest that a combination of silencing factors, such as DNMT3A/DNMT3L and KRAB, are required to fully heterochromatinize a given promoter and lead to permanent silencing of a gene [44, 80].

## Conclusion

This work constitutes a broad assessment of the transcriptional responses and heritability of these responses upon promoter DNA methylation in the human genome. In the future, it will be important to undertake comprehensive epigenome manipulation in other cell types and states in order to establish the generalizability of these relationships, most importantly in pluripotent and distinct differentiated cell types. Our work demonstrates the utility of artificial epigenome editing to understand the information content of covalent DNA modifications and highlights new challenges that need to be overcome for their effective use.

## Methods

### Lentiviral constructs, MCF-7 cell lines, and FACS sorting

The construct named ZF-D3A-wt (Additional file 1: Fig. S1A) is under an inducible Dox promoter. Downstream of the zinc finger and the DNMT3A catalytic domain, we included a HA-epitope to be able to perform chromatin immunoprecipitation and a self-cleaving peptide (P2A) plus a green fluorescent protein (GFP) to be able to purify ZF-D3A-wt expressing cells by FACS (Fig. 1A). In parallel, we generated another identical version of this construct harboring 4 amino acid mutations (F636A, E660A, E725A, R295A) on the methyltransferase domain (ZF-D3A-mut), which are known to abrogate the capacity to methylate cytosines [58] (Fig. 1A, Additional file 1: Fig. S1A). Using lentiviral vectors, we generated MCF-7 Dox-inducible cell lines for ZF-D3A-wt and ZF-D3A-mut. For lentivirus production, 3 to 4 million HEK293T cells were seeded in 10-cm plates and cultured in 10-ml complete media (DMEM supplemented with 10% FBS and 1X (v/v) Glutamax) 24 h prior to transfection. The next day, media were removed and replaced with 10 ml of fresh complete media containing 25 μM chloroquine and incubated for 1 h. Transient transfection was carried out by mixing plasmids encoding lentiviral envelope pMD2G (#12259, Addgene) and lentiviral packaging psPAX2 (#12260, Addgene) with lentiviral vector encoding either ZF-D3A-wt or ZF-D3A-mut in the following ratio 3.5 μg:7.5 μg:10 μg in 700 μl of sterile 0.25 M $CaCl_2$. The solution was vortexed thoroughly to mix. Next, the plasmid and $CaCl_2$ solution was added dropwise in a new 15-ml falcon tube containing 700 μl of 2X HBS buffer (274 mM NaCl, 10 mM KCl, 1.4 mM $Na_2HPO_4$, 15 mM D-glucose, 42 mM HEPES, pH 7.05) while the solution was continuously bubbled using an automatic pipette pump attached to a 1-ml serological pipette to blow air in the solution. The mixture was incubated at room temperature for 30 min to allow calcium phosphate precipitation to form. Transfection mixture was then added dropwise to HEK293T cells and was mixed by swirling the plate gently before returning to the incubator. Lentiviral particles were harvested 48 h later and were filtered using 0.45-μm filters. For transduction, 3–4 million of MCF-7 were seeded in 10-cm plates and cultured in 10-ml complete media 24 h before transduction. The next day, the media were removed and replaced with 5-ml complete media and 5-ml media containing lentivirus. A total of 4 μg/ml polybrene was added to increase transfection efficiency. Cells were incubated for 48 h to express the transgene before being selected with 2 μg/ml puromycin and 800 μg/ml G418 for 14 days to ensure stable incorporation of the transgenes. For Dox induction, cells were seeded 24 h prior to adding Dox (Clontech) to the complete media. We tested two different concentrations of Dox 100 ng/ml and 1000 ng/ml for 3 and 6 days for optimal methylation induction (Additional file 1: Fig. S12), with day 3 1000 ng/μL reaching the highest induction. For all downstream sequencing experiments, ZF-D3A-wt and ZF-D3A-mut stable cells were collected at various stages: without Dox in the media (noDox), after 3 days of 1000 ng/ml Dox induction (Dox), and upon 7 days of Dox withdrawal (DoxWD) after sorting for GFP expressing cells (Fig. 1B). Additionally, ATAC-seq and WGBS samples were collected for 3 days post-Dox, also after sorting for GFP expressing cells.

The MCF-7 cells were not authenticated.

### ZF-D3A and H3 modification ChIP-seq

Chromatin immunoprecipitation followed by high-throughput DNA sequencing (ChIP-seq) for the HA-tag (for HA-tagged ZF-D3A localization), as well as for the H3K4me3, H3K27me3, and H3K9me3 histone modifications, was performed as described previously [39]. Briefly, cells were crosslinked for 10 min in 1% formaldehyde and quenched in 125 mM glycine. Prior to ChIP, antibodies were bound to beads by mixing 4 µg of either HA.11 antibody (#901502, Biolegend), H3K4me3 (#C15410003, Diagenode), H3K9me3 (# ab8898, Abcam), or H3K27me3 antibody (ab6002, Abcam) with 50 µl washed Dynabead Protein G (#10003D, Thermo Fisher Scientific) in 500 µl RIPA-150 buffer (50 mM Tris-HCl pH 8.0, 0.15 M NaCl, 1 mM EDTA , 0.1% SDS, 1% Triton X-100, and 0.1% sodium deoxycholate) and incubated at 4℃ for 6 h on a rotator. Crosslinked cells were lysed on ice for 10 min in 15 ml ChIP lysis buffer (50 mM HEPES pH 7.9, 140 mM NaCl, 1 mM EDTA, 10% glycerol, 0.5% NP-40, 0.25% Triton X-100) supplemented with 1x EDTA-free Protease Inhibitor Cocktail (#11836170001, Sigma-Aldrich). Lysed cells were centrifuged at 3200×g for 5 min, supernatant removed and followed by two washes with 10 ml ChIP wash buffer (10 mM Tris-Cl pH 8.0, 200 mM NaCl, and 1 mM EDTA pH 8.0). Lysed cells were resuspended in 130 µl nuclei lysis buffer (50 mM Tris-HCl pH 8.0, 10 mM EDTA, and 1% SDS) supplemented with 1x EDTA-free Protease Inhibitor Cocktail (#11836170001, Sigma-Aldrich), transferred to Covaris tubes (#520045, micro-TUBE AFA Fiber 6x16mm), and sheared with the Covaris (S220) for 5 min (5% duty cycle, 200 cycles per burst, and 140 watts peak output at 4℃). Sheared chromatin was transferred to 1.5-ml Eppendorf tubes, centrifuged at 10,000×g for 10 min. The supernatant was transferred to 2-ml low-bind tubes (#AM12475, Thermo Fisher Scientific) containing 1.2 ml ChIP dilution Buffer (50 mM Tris-HCl pH 8.0, 0.167 M NaCl, 1.1% Triton X-100 and 0.11% sodium deoxycholate) and 0.65 ml RIPA-150 buffer and incubated with the previously prepared antibody-bound Dynabeads at 4℃ overnight on a rotator. Chromatin-bound beads were subsequently washed one time with 1 ml RIPA-150 buffer, two times with 1 ml RIPA-500 buffer (50 mM Tris-HCl pH 8.0, 0.5 M NaCl, 1 mM EDTA, 0.1% SDS, 1% Triton X-100, and 0.1% sodium deoxycholate), two times with 1ml RIPA-LiCl buffer (50 mM Tris-HCl pH 8.0, 1 mM EDTA, 1% NP-40, 0.7% sodium deoxycholate, and 0.5 M LiCl$_2$), and two times with TE buffer (10 mM Tris-HCl, pH 8.0, 0.1 mM EDTA). After wash steps, DNA was eluted, crosslinks were reversed, and immunoprecipitated DNA was purified with 2x (v/v) Agencourt AMPure XP beads. ChIP-seq libraries were prepared from ChIP eluate containing 10 ng DNA using the SMARTer ThruPLEX DNA-Seq Kit (Takara, R400675) with SMARTer DNA unique dual index (Takara, R400665). After limited PCR amplification, libraries were purified with 1.1x (v/v) Agencourt AMPure XP beads and eluted in a final volume of 20 µl. Libraries were then sequenced with a NovaSeq 6000 (Illumina).

Multiplexed library sequencing yielded 24–38M pair reads per experiment for the ZF-D3A HA experiments and ~30–50M single-end reads for the H3 modification ChIP-seqs.

ChIP-seq reads were trimmed with bbduk [81] and mapped to the genome using Bowtie 2 [82] allowing for an insert site of 2000 bp, reads were deduplicated using Sambamba markdup function [83], and reads mapping to blacklisted genomic regions were filtered out using SAMtools [84]. We then used MACS2 to call peaks with the matched

chromatin input sample as background for each ZF-D3A HA ChIP-seq experiment, specifying the parameters "-f BAMPE -q 0.05 --down-sample" [85]. The peaks for replicate pairs were merged using IDR (https://github.com/nboley/idr) [86]. Background subtracted BigWig tracks were obtained with deepTools, using CPM normalized tracks for ZF-D3A ChIP and input [87].

Histone modification and TF MCF-7 ChIP-seq coverage tracks mapped to the hg19 genome were downloaded from the ENCODE portal [88], specifically ENCFF245TUO, ENCFF711PMS, ENCFF635LQF, ENCFF561UXI, ENCFF000QRT, ENCFF249SMB, and ENCFF099HRD.

### ATAC-seq

ATAC-seq was performed following the Omni-ATAC protocol [89]. Briefly, 50,000 FACS-purified cells were resuspended in 50 µl cold ATAC resuspension buffer 1 (10 mM Tris-HCl pH 7.4, 10 mM NaCl, 3 mM $MgCl_2$, 0.1% (v/v) NP40, 0.1% (v/v) Tween-20, 0.01% (v/v) Digitonin) and incubated on ice for 3 min to lyse cells. The lysis reaction was stopped by adding 1 ml of cold ATAC resuspension buffer 2 (10 mM Tris-HCl pH 7.5, 10 mM NaCl, 3 mM $MgCl_2$, 0.1% (v/v) Tween-20) and inverting the tube for 3 times to mix. The nuclei were pelleted by centrifugation at $500 \times g$ for 10 min at 4°C in a fixed angle centrifuge. The supernatant was removed and the nuclei were resuspended in transposition mix containing 25 µl of 2x TD buffer (20 mM Tris-HCl pH 7.5, 10 mM $MgCl_2$, 20% (v/v) Dimethyl Formamide), 2.5 µl of 100 nM Tn5, 16.5 µl PBS, 0.5 µl of 1% (v/v) digitonin, 0.5 µl of 10% (v/v) Tween-20, and 5 µl $H_2O$. The transposition reaction was incubated at 55°C for 30 min in a thermomixer with 1000 RPM mixing. Transposed DNA was cleaned up using the Qiagen MinElute PCR Purification kit following manufacturer's instructions. Twenty microliters of eluted DNA was subject to 8–10 cycles of PCR amplifications using NEBNext 2x MasterMix. Multiplexed libraries were then sequenced using an Illumina NovaSeq 6000, obtaining an average of 90 M paired-end reads per sample.

Sequenced ATAC-seq reads were then trimmed with bbduk for Nextera adapters and mapped to the genome using Bowtie 2. Duplicated reads were removed and blacklisted regions filtered as with ChIP-seq, and peaks were called using MACS2 with "--nomodel -f BAM --keep-dup all" parameters. Replicates were merged using IDR, and each sample was validated visualizing the insert size distribution following nucleosome periodicity [90]. To obtain the nucleosome positional information, we merged replicates for each condition and ran nucleoATAC with default parameters on a combination of all peaks in all samples (+2000 bp upstream/downstream) [61]. To identify peaks with differential accessibility, we used the merged ATAC peak file for all samples and counted reads overlapping with peaks in each sample with the R GenomicAlignments package [91], which were then analyzed with DESeq2 [92].

We obtained TF footprints using the merged replicate bam files for noDox, Dox, and Dox-mut ATAC-seq experiments. We first corrected the Tn5 insertion bias with TOBIAS [93] and assigned footprints to TF using the jaspar vertebrate motif database (JASPAR2020_CORE_vertebrates_non-redundant_pfms_jaspar.txt). We calculated differential TF footprinting with the BINDETECT function, using different sets of regions

(promoter-DMRs or merged ATAC peaks) in each comparison. Similarity across ATAC-seq replicates was assessed using DeepTools2 (Additional file 1: Fig. S13).

### RNA-seq

Fifty thousand FACS-purified cells were subject to RNA purification using the RNAdvance Cell v2 kit following manufacturer's instructions. Cells were lysed using the lysis buffer from the kit and mixed with 2 µl of 1:100 diluted ERCC RNA spike-in mix 1 or mix 2 (ThermoFisher Scientific, #4456739) prior to RNA extraction. Total RNA with spike-in was subjected to library generation using the Illumina TruSeq Stranded mRNA kit according to the manufacturer's instructions, except one-third of all reaction volumes were used and the final amplification used 13 cycles of PCR. The RNA-seq libraries were multiplexed on a NovaSeq run, obtaining an average of 80 million paired-reads per replicate. The reads were trimmed using fastp with default parameters [94], then mapped with HISAT2 (--rna-strandness RF) to an index including hg19 genome assembly and ERCC spike-in sequences [95]. Then, RNA-seq read-pairs were read into R using summarizeOverlaps function within the GenomicAlignments package and mapped to the UCSC hg19 gtf [91]. The count matrix was then processed with DESeq2 [92] after filtering out non-expressed genes and very lowly expressed genes (>0 counts in ≥10 samples). Samples were normalized using size factors for each pairwise condition comparisons. In parallel, RNA-seq was mapped to the reference human transcriptome including the ZF-D3A sequence with Kallisto [96].

### Whole genome bisulfite sequencing (WGBS) and Tet-assisted bisulfite sequencing (TAB-seq)

DNA was extracted from FACS-purified cells using the Qiagen DNeasy Blood & Tissue Kit following the manufacturer's instructions. A total of 500 ng of genomic DNA was spiked with 0.5% (w/w) of unmethylated lambda phage DNA (Promega) for calculation of the bisulfite non-conversion rate and sheared with a Covaris S2 sonicator to an average length of 300 bp. The sheared DNA was end-repaired, A-tailed, and ligated to methylated adapters (NEXTflex Bisulfite-Seq Barcodes, PerkinElmer) using the NxSeq AmpFREE Low DNA Library Kit (Lucigen). Adapter-ligated libraries were subjected to bisulfite conversion using the EZ DNA Methylation-Direct Kit (Zymo) following the manufacturer's instructions and subjected to 6 cycles of PCR amplification using KAPA HiFi Uracil+ DNA polymerase (KAPA Biosystems).

The WGBS libraries were then sequenced with a NovaSeq 6000 (Illumina) instrument obtaining ~500M million paired-end 112-bp reads per ZF-D3A-wt samples (~40x) and ~200M million for ZF-D3A-mut samples and DoxWD-3d (~14x). The TAB-seq sample was sequenced on a HiSeq 1500 (Illumina), obtaining 800 M single-end 100-bp reads.

The sequenced reads were first trimmed using bbduk, and overlapping pairs were merged using BBMerge [97]. The reads were mapped to the hg19 genome including the lambda and pUC19 sequences using BS-Seeker2 with Bowtie 2 as back-end aligner (-e 300 -X 2000) [98]. The duplicated reads were removed using Sambamba

markdup for paired-end reads and PALEOMIX for merged reads [99]. We then used CGmapTools to generate methylation calls [100].

For TAB-seq, genomic DNA was isolated as described for WGBS for a MCF-7 cell line lacking the GFP on the ZF-D3A construct. TAB-seq libraries were generated using the 5hmC TAB-seq kit (WiseGene) according to the manufacturer's instructions [76]. 5-hydroxymethylated pUC19 DNA (WiseGene) was used to estimate the protection of 5hmC by β-glucosyltransferase, and unmethylated lambda phage DNA was used to estimate the bisulfite non-conversion rate. Single-end 100 cycle sequencing was performed on a HiSeq1500. Reads were mapped as described for WGBS.

### ChIP-bisulfite-sequencing

Two 15-cm plates of MCF-7 cells encoding a version of the ZF-D3A without the GFP were grown and doxycycline induced for 3 days. Cells were washed two times with 10 ml of PBS and crosslinked for 5 min in 50 mM HEPES-KOH, pH 7.5, 100 mM NaCl, 1mM EDTA, and 1% formaldehyde. The crosslinking reaction was quenched by the addition of glycine to a final concentration of 125 mM and washed twice with phosphate-buffered saline (PBS). All subsequent solutions were supplemented with a protease inhibitor cocktail (Sigma Cat. # P8340). Cells were scraped off the plates with a rubber policeman in 10 ml of PBS and centrifuged at 3000 rpm for 5 min in a swinging bucket rotor. The cell pellets were resuspended in 10 ml of 50 mM HEPES-KOH, pH 7.9, 140 mM NaCl, 1 mM EDTA, 10% glycerol, 0.5% NP-40, and 0.25% Triton X-100; incubated on ice for 10 min; and centrifuged at 3000 rpm for 10 min in a swinging bucket rotor. Cell pellets were washed twice by gently adding 10 mM Tris-HCl, pH 8.1, 200 mM NaCl, and 1 mM EDTA to the cell pellets, trying not to disturb the pellets, and centrifuged at 3000 rpm for 5 min. Finally, the cell pellets were resuspended in 0.1% SDS and 1 mM EDTA and transferred to a Covaris TC12x12 tube. The chromatin was sheared using a Covaris S2 sonicator with the following settings: time 12 min, duty cycle 5%, intensity 4, cycles per burst 200, temperature 4°C, power mode frequency sweeping. Triton X-100 and NaCl were added to a final concentration of 1% and 150 mM respectively. The sheared chromatin was centrifuged at maximum speed in a microfuge for 15 min at 4°C, and the supernatant was transferred to a new tube. Two microliters of anti-H3K4me3 (Diagenode, Cat. # C15410003) or 4 μl of anti-phospho-Ser5 RNA polymerase II antibody (Active Motif, Cat. # 39233) was added and incubated overnight at 4°C. Thirty microliters of Protein G Dynabeads (Life Technologies) was added and incubated on a tube rotator for 90 min at 4°C. The beads were washed twice with 20 mM HEPES-KOH, pH 7.9, 0.1% SDS, 150 mM NaCl, 1% Triton X-100 2 mM EDTA; twice with 20 mM HEPES-KOH, pH 7.9, 0.1% SDS, 500 mM NaCl, 1% Triton X-100 2 mM EDTA; once with 100 mM Tris-HCl pH 7.5, 0.5 M LiCl, 1% NP-40, 1% sodium deoxycholate; and once with 10 mM Tris-HCl, pH 8.0, 1 mM EDTA. The DNA was eluted twice by incubating for 30 min in 25 μl of 20 mM HEPES-KOH, pH 7.9, 1 mM EDTA, 0.5% SDS, and 0.5 mg/ml Proteinase K. To the 50 μl of eluted DNA, 3 μl of 3M sodium acetate, pH 5.3, and 0.5 μl 30 mg/ml RNase A were added and incubated overnight at 65°C in a hybridization oven. 1.5 μl of 20 mg/ml proteinase K was added and incubated for 1 h at 50°C and the DNA was purified with 2 volumes of SPRI beads and eluted in 20 μl Tris-HCl, pH 8.0, 0.1 mM EDTA. Libraries were made with the Accel-NGS Methyl-Seq DNA Library Kit (Swift Biosciences) according

to the manufacturer's instructions. Reads were aligned and DNA methylation sites were identified as described for WGBS. Peaks were called as for ChIP-seq datasets.

### Targeted bisulfite-PCR amplicon sequencing

DNA was extracted from FACS-sorted cells using the Qiagen DNeasy Blood & Tissue Kit and 500 ng of genomic DNA was spiked with 0.5% (w/w) of unmethylated lambda phage DNA (Promega) for calculation of the bisulfite non-conversion rate and bisulfite converted with the EZ DNA Methylation-Direct Kit (Zymo Research). PCR amplicons were designed with methprimer and bisulfite-converted DNA was amplified with 40 cycles of PCR using MyTaq HS mix (Bioline). PCR reactions were pooled and purified with SPRI beads. One microgram of pooled PCR products in 14.5 μl were phosphorylated by adding 15 μl 2X Quick Ligase Buffer (New England Biolabs) and 0.5 μl T4 polynucleotide kinase (New England Biolabs) and incubating at 37°C for 30 min. Illumina TruSeq adapters synthesized by IDT and annealed by heating to 99°C and slowly cooling to 20°C were ligated to the phosphorylated PCR products by adding 3.75 μl 10 μM annealed TruSeq adapters, 10 μl 2X Quick Ligase Buffer (New England Biolabs), and 6 μl water. The ligation reactions were incubated at 25°C for 20 min and stopped by adding 2 μl 0.5 M EDTA. The DNA was purified by adding 20.8 μl (0.4 volumes) of SPRI beads. The libraries were subjected to single-end 300 cycle sequencing on the Illumina MiSeq.

### Methylation analysis

Methylation levels at CpGs were read into R using the bsseq package, and strands were collapsed using the strandCollapse function [101]. Lambda genome data was extracted to calculate the bisulfite non-conversion rates for each library (Additional file 2: Table S1).

We called DMRs across pairwise comparisons using 2 replicates for each condition and the dmrseq package (bpSpan = 500, maxGap = 500, maxPerms = 20) [102]. These DMRs were initially filtered for a *q*-value < 0.05 and a minimal weighted mCG difference ≥ 20%. For the promoter-DMR analysis, we only kept the more stringent subset of DMRs with *q*-value < 0.01 overlapping −2000/+200 bp of any TSS. The analysis represented in Fig. 4 is promoter-DMRs for genes that are not differentially expressed in the ZF-D3A-mut Dox vs noDox. UMRs were obtained using MethylSeekR [103]. Cross-sample methylation clustering was obtained using weighted average mCG levels on UMRs and LMRs obtained from the noDox MethylSeekR run (Additional file 1: Fig. S13B). DMRs were then classified as retain-DMRs if they had <10% mCG in noDox, >10% mCG in Dox and DoxWD-7d, and at least twice as much mCG in DoxWD-7d compared to noDox. Loss-DMRs were required to have mCG <10% in noDox and DoxWD-7d but >10% mCG in Dox.

The analysis of single WGBS reads (Fig. 4e) was performed by selecting reads for which 80% of their sequence overlapped with promoter-DMRs. Only reads harboring ≥5 CpGs were selected.

### Alternative promoter analysis

BAM files generated from RNA-seq mapping were subject to differential alternative promoter usage calling between noDox vs Dox and noDox-mut vs Dox-mut using

SEASTAR [104] with the denovo mode. Alternative TSS data are found in Additional file 2: Tables S3 and S4.

### Motif analysis and data visualization

We used HOMER2 to obtain de novo motifs in the ZF-D3A-wt ChIP-seq merged peaks [105]. The motif scans were also performed with Homer2, using the findMotifsGenome. pl function. For promoter-DMRs, the background used was the rest of promoter-DMRs (e.g., downregulated promoter-DMRs vs upregulated + non-differentially expressed). Similarly, background sequences for ATAC peaks that gain accessibility were peaks that lose accessibility, and retain-DMRs were used as background for loss-DMRs and vice versa. Bonferroni-corrected $p$-values from HOMER2 were used to filter out motifs with $p < 0.01$. Additionally, we required 10% of the target sequences having the motif and a minimal fold enrichment between target and background > log2(0.3). scanMotifGenomeWide.pl was used to find the ZF-D3A motif across the genome.

Data visualization in heatmaps and average plots were obtained using the deepTools package, and genome visualizations were obtained using IGV genome browser. There is a public genome browser session to visualize this data at http://tinyurl.com/sc6jchg.

### Supplementary Information

---

Additional file 1: Figure S1. The Dox inducible ZF-D3A system. Figure S2. ZF-D3A ChIP-seq replicates show equiparable positional enrichments. Figure S3. Nucleosome position dictates mCH deposition on closed and open chromatin. Figure S4. ZF-D3A-wt expression does not lead to a global reduction of transcriptional activity. Figure S5. ZF binding and DMR induction led to diverse transcriptional responses. Figure S6. Characterisation of promoter DMRs. Figure S7. Examples of non-responsive promoter-DMRs. Figure S8. Differential TF-footprinting at promoter-DMRs. Figure S9. Alternative TSS usage upon Dox induction. Figure S10. Differential ATAC-seq peaks. Figure S11. Differential methylation retention at promoters and distal regulatory elements. Figure S12. Prolonged Dox induction does not increase methylation gain. Figure S13. Epigenomic data overview. Table S1. Global WGBS stats for all samples. Table S2. List of promoters with DMRs and transcriptional change. Table S3. Alternative TSS in noDox vs Dox comparison. Table S4. Alternative TSS in noDox-mut vs Dox-mut comparison.

Additional file 2: Supplementary Table S1: WGBS stats, S2: Promoter DMRs, S3: Alternative TSS in ZF-D3A-wt, and S4: Alternative TSS in ZF-D3A-mut.

Additional file 3. Review history.

---

#### Acknowledgements

We thank C. Pflüger, M. Cruickshank, and M. Lewsey for critical reading of this manuscript; Andrea Holme at the UWA Centre for microscopy, characterization, and analysis; and Ji Kevin Li at the Harry Perkins Institute of Medical Research Flow Cytometry Facility for performing the FACS and flow cytometry analysis. We also thank Rafael Irizarry and Keegan Korthauer for releasing their analysis code.

#### Peer review information

#### Review history

The review history is available as Additional file 3.

#### Authors' contributions

RL and PB conceived the study. RL, EF, TVN, and AdM designed the study. EF and TVN performed experiments with the help of OB and SS. AdM, EF, and TVN performed analysis with the help of SB. EF, DP, and JP performed the sequencing. PJF, SS, PB, and MRG provided materials. AO contributed statistical analysis design. EF, RL, and AdM wrote the manuscript, and all authors read and commented on the manuscript. The authors read and approved the final manuscript.

#### Authors' information

Twitter handles: @deMendoza_Alex (Alex de Mendoza); @trungbiotech (Trung Viet Nguyen); @ry_lister (Ryan Lister).

#### Funding

This work was funded by the Australian National Health and Medical Research Council (GNT1069830), the Australian Research Council (ARC) Centre of Excellence program in Plant Energy Biology (CE140100008), the National Institutes

of Health (R01CA170370 and 1R01DA036906), the Raine Medical Research Foundation, the National Cancer Institute (P30CA014089), and the National Human Genome Research Institute (R21HG006761). RL was supported by an ARC Future Fellowship (FT120100862), a Sylvia and Charles Viertel Senior Medical Research Fellowship, a Howard Hughes Medical Institute International Research Scholarship, and an NHMRC Investigator Grant (GNT1178460). OB was supported by an Australian Research Council Discovery Early Career Researcher Award (DE140101962). AdM was supported by an EMBO Long Term Fellowship (ALTF144-2014). PB was supported by an ARC Future Fellowship (FT130101767) and by a Cancer Council Western Australia Research fellowship. TVN was supported by the Forrest Research Foundation PhD scholarship. Genomic data was generated at the Australian Cancer Research Foundation Centre for Advanced Cancer Genomics and at the LIEF (LE110100188) funded sequencing facility.

### Availability of data and materials

The sequencing data have been deposited in the Gene Expression Omnibus under the accession numbers GSE165891 [106] and GSE102395 [107]. Code and materials are available from GitHub [108].

## Declarations

### Ethics approval and consent to participate

Not applicable.

### Consent for publication

Not applicable.

### Competing interests

The authors declare that they have no competing interests.

### Author details

[1]Australian Research Council Centre of Excellence in Plant Energy Biology, School of Molecular Sciences, The University of Western Australia, Perth, WA 6009, Australia. [2]Harry Perkins Institute of Medical Research, QEII Medical Centre and Centre for Medical Research, The University of Western Australia, Perth, WA 6009, Australia. [3]School of Biological and Behavioural Sciences, Queen Mary University of London, Mile End Road, London E1 4NS, UK. [4]Department of Biochemistry and Molecular Medicine, University of Southern California, 1450 Biggy St, Los Angeles, CA 90089, USA. [5]Integrated Genetics and Genomics, University of California, Davis, 451 Health Sciences Dr, Davis, CA 95616, USA. [6]Department of Neurological Surgery, University of California, 1450 3rd St, San Francisco, CA 94158, USA. [7]School of Anatomy, Physiology and Human Biology, The University of Western Australia, 35 Stirling Hwy, Crawley, WA 6009, Australia. [8]Genomics and Epigenetics Division, Garvan Institute of Medical Research, Sydney, New South Wales, Australia. [9]School of Biotechnology and Biomolecular Sciences, University of New South Wales, Sydney, New South Wales 2052, Australia. [10]The Peter MacCallum Cancer Centre, 305 Grattan St, Melbourne, VIC 3000, Australia. [11]School of BioScience, The University of Melbourne, Parkville, VIC 3010, Australia. [12]The Greehey Children's Cancer Research Institute, The University of Texas Health Science Center at San Antonio, San Antonio, TX, USA.

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

## 