## [Additional file 3. Review history. · Genome Biology]

Review History

First round of review

Reviewer 1

Are you able to assess all statistics in the manuscript, including the appropriateness of statistical tests used? No, I do not feel adequately qualified to assess the statistics.

Comments to author:

Professor Ryan Lister and colleagues' manuscript entitled "Limited repressive capacity of promoter DNA methylation revealed through epigenome manipulation" exhibits a very comprehensive investigation regarding the repressive effect of DNA methylation editing in promoter regions in human cells. By applying a spectrum of high-throughput methods, this manuscript constitutes a broad assessment on the relationship between forced DNA methylation and transcriptome, histone modification, and chromatin accessibility.

Although an evident path that bridges DNA methylation editing in promoter and the manipulation of gene repression has not been guaranteed in this manuscript, multiple critical implications have been provided herein, like the list of hypermethylating sensitive promoters, promoter switch in DNA methylation editing, involvement of TETs in differential methylation retention, et. al. Nevertheless, it is believed that following revisions are essential to fit Genome Biology's requirement.

1) The simultaneously multiply (thousands) DNA methylation editing relies on a single zinc-finger-DNMT3A fusion protein. Multi-targeting capability is mainly based on the relatively broad specificity of the Sox2 zinc finger. Among the targeted sites, only ~36% encode the 12 bp core sequence, which rises the concern that genome-wide DNA methylation gain may be irrelevant to the limited specificity provided by ZF. More DNA methylation editing profiles and the distribution of ZF-DNMT3A binding peaks data from a few additional ZFs-DNMT3A fusion proteins could solidly support authors' conclusion.

2) Dox withdraw (DoxWD) experiment provides the valuable views to decipher the retention and loss of DNA methylation marks after editing. However, how the DoxWD samples obtained is not clearly displayed in manuscript. I may miss the method detail of this experimental design, thus it will be helpful if authors could confirm that the Dox withdraw was from the cell population 1) GFP+ sorted cells or 2) the bulk cells (both GFP+ and GFP- after 3-day induction). The decrease of GFP signal may come from the withdraw of Dox induction or the lethal cytotoxicity from the global hypermethylation. The latter situation may partially explain the result why 6-day Dox induction achieved lower methylation level than 3-day induction (Line 766).

3) The intermediate status between Dox and DoxWD (day 7) is critical to figure out the role of TET-mediated demethylation in controlling the rates of methylation loss. An additional time point at day 2 or day 3 post the Dox withdraw will provide the insight regarding correlation between the status of TF binding/chromatin accessibility and the rate of methylation loss.

- 4) The manuscript focuses on the study of the repressive effect from promoter methylation editing. However, only H3K4me3 mark profiles are included. The repressive histone modification marks like H3K9me3 and H3K27me3 are not mentioned in current manuscript.
- 5) Extra discussion regarding the difference of ChIP-seq mapping signals between ZF-D3A-wt and ZF-D3A-mut enrichment (Figure S2B) should be provided. More than a half of peaks are ZF-D3A-wt only is an amazing discovery.
- 6) Data details or interpretation should not be included in Method section (Line 766 - 771).
- 7) Current typesetting needs further polishing. The full name of abbreviation terms (like "CTD" in Line 559; "mC" in Line 545, et.al.) are not included, and typo errors (like "reconFig. DNA" in line 635, et.al.) should be revised.

Reviewer 2

Are you able to assess all statistics in the manuscript, including the appropriateness of statistical tests used? Yes, and I have assessed the statistics in my report.

Comments to author:

This manuscript from de Mendoza et al. describes a fascinating set of -omics experiments where they broadly target a zinc-finger methyltransferase across the genome of MCF7 cells and analyze the resultant effects on DNA methylation, chromatin structure, and expression. There has been much recent interest in epigenome editing as an effective measure of the downstream functional effects of methylation changes as well as a potential therapeutic. However, our understanding of how to control this approach and how the genome constrains this technology is still fairly limited. This work would thus help in the future design of epigenome targeting experiments. The work is quite rigorous, and the positive conclusions (e.g. that sometimes methylation can act as a repressor or activator depending on what TF is affected) are important and well supported. However, there are some broader negative conclusions about methylation-mediated promoter silencing that are not well supported since it's difficult to prove a negative conclusion, especially without substantial more work. For example, it's very hard to sort out the effects at individual promoters when large numbers of genes changed, including potential silencing or activation of other cis-acting factors. Further it's difficult to rigorously assess that the level of induced methylation and the sites methylated were sufficient to induce methylation as would be expected by the standard hypothesis that promoter methylation silences expression. I do wonder how much of what they learned here would transfer to CRISPR-based editing systems. These systems are potentially more impactful since they are cheaper and easier to implement. That said, there is truly an impressive assortment of -omics analysis to support their observations. Many of the major points about how induced methylation may silence or activate gene expression depending on the context are under appreciated and this work establishes that global trends based on these mechanisms are observed.

Major Comments

- (1) Some conclusions are overstated, especially concerning how promoter methylation may not silence expression. While they very well may be right, this is hard to prove, and the data presented do not go far enough to support these broad statements. For example:
 - a. In the abstract: "methylation is often not sufficient to suppress transcription" - this would take a lot more work to show that unsilenced promoters were sufficiently methylated (most were only methylated at < 40%) across the entire promoter or CGI and in multiple contexts. This is not sufficiently tested. Instead,

there is a beautiful demonstration that methylation can have a variety of context specific effects (e.g. silencing, activation, repression)

b. In the abstract: "in a process combining active and passive demethylation". - just because it is possibly TET mediated does not mean it's not also cell division dependent, which usually falls under "passive demethylation" rather than active. Better to say something like passive and TET-mediated demethylation. There is a similar problem with the conclusion on page 28 line 625.

c. In the conclusions, "this work constitutes a broad assessment of the repressive capacity of promoter DNA methylation". Again, this isn't really true, since it's hard without substantial more work to conclude that every promoter was induced sufficiently to test its repressive capacity. The title uses similar problematic language and in this way it's not really indicative of what the major part of the work is about.

(2) In the TET analysis, how significant is the small amount of hydroxymethylation? It appears this is ~2% from TAB-seq data. Is this found at ~2% across all retain and loss DMRs, or is it much higher at a subset DMRs and near 0 at others at much higher levels? 2% is found at retain and loss DMRs equally, what about non-DMR promoters? I.e. is 2% just a general background rate. Is there evidence that 2% at a promoter is indicative of methylation turnover? Or just a minor inconsequential background rate.

(3) I was surprised in the motif search where the authors identify potential binding sites (p. 9 and Fig 2B) that they chose a single consensus motif rather than running a FIMO or similar search with the entire position specific weight matrix. Do the results change substantially when you open it up to all potentially bound sequences?

(4) On page 30, line 674. Are these different features true of any ZF-D3A construct? Or are they specific to just this one? For example, it seems this construct was designed to a CGI, so if you didn't design to a CGI would the off targets still be at high-CpG density promoters?

(5) I think there are some caveats to their heterogeneity analysis on p. 17 and 18. First reads are not long enough to consider a full +/-2kb region across a broad promoter window (I think that's what they used to call promoter DMRs). For example, the fully methylated reads could be coming from a section of the promoter whose methylation has no bearing on gene silencing, while a large different section does with intermediate methylation. Second, they made large numbers of modifications across the genome at different loci, it is possible for some cells to lose repressors or gain activators or alternative promoters and others to not. This could dampen transcriptional readouts as well. Further complicating matters, MCF7 cells are a heterogeneous mix, even before you start hitting them with different constructs.

(6) In the discussion there is mention that DNMT3A + 3L + KRAB was shown to be sufficient for longer term silencing. I also thought there has been some discussion in the literature of the need to hit loci harder (e.g. multiple DOX inductions or with SunTag system) and that these could stabilize the methylation even without KRAB recruitment.

(7) Since CRISPR-based epigenome editing is more common (since it's substantially easier and cheaper) than zinc-finger based systems, it would be good if the discussion could talk more about which of their findings will likely hold up in these other systems based on current literature, and which are potentially specific to the zinc finger system.

(8) There is a fundamental limitation to this work in that it is only performed in a single cell line. The authors acknowledge this limitation and personally I find this work quite compelling. However, my guess is critics will use this as a dismissive comment. This is merely a comment to the authors. I certainly do not expect them to repeat their findings in another line at this stage. But I did want to mention it in case it is helpful. One possible way to help deal with such critiques would be when they highlight how different observations are supported in the literature in the results, they could add a quick comment the cell line or

context.

Minor Comments

(a) p. 3, line 72 "5' position" should be "5 position".

(b) p. 9 line 231 states that the data indicates "that the ZF-D3A binding capabilities are similar to those of a pioneer TF." And then later on p. 11 they seem to conclude the opposite. On p. 9 they might be better of stating that this data suggests that ZF-D3A could act as a pioneering factor, so we decided to test this using the following analysis. Or something like that to avoid confusion of conflicting conclusion-like statements.

(c) p. 19 line 422, comments about Sox2 lacking methyl-sensitivity is missing a reference.

(d) p. 28 line 635, reconFig. Should be reconfigure.

(e) The wording on p. 30 line 678 seems off. Normally nucleosomes facilitate D3A-dependent methylation deposition. The "even using" construct implies that for both their fusion and in normal function nucleosomes impede methylation. I think the wording just needs to be adjusted a little.

Reviewer 3

Are you able to assess all statistics in the manuscript, including the appropriateness of statistical tests used? Yes, and I have assessed the statistics in my report.

Comments to author:

I've read this updated manuscript with great interest. It is a vastly improved version of the manuscript that was on bioRxiv already since 2017. I have to say that this is excellent work, highly important for the field and for understanding the epigenetic regulation and the role DNA methylation plays. The study is very well designed and controlled, while taking into consideration many alternative explanations it clearly shows that DNA methylation is not sufficient to provide gene repression and that additional factors also play a role. Whereas for the initial bioRxiv manuscript, I had doubts to some parts and the data could be explained otherwise, it seems that the authors managed to very comprehensively address them in this submitted version.

Dear Dr. Kevin Pang,

Thank you for taking the time to review our previous submission by de Mendoza, Nguyen, Ford *et al.* entitled "*Limited repressive capacity of promoter DNA methylation revealed through large-scale epigenome manipulation*". Herein we are re-submitting a revised version of this manuscript addressing the changes requested by the reviewers. We would like to first thank you and the three reviewers for the constructive feedback and time and effort in considering our study for publication.

Below we provide a point by point response to the Reviewers comments and suggestions.

REVIEWER COMMENTS

Reviewer #1:

Professor Ryan Lister and colleagues' manuscript entitled "Limited repressive capacity of promoter DNA methylation revealed through epigenome manipulation" exhibits a very comprehensive investigation regarding the repressive effect of DNA methylation editing in promoter regions in human cells. By applying a spectrum of high-throughput methods, this manuscript constitutes a broad assessment on the relationship between forced DNA methylation and transcriptome, histone modification, and chromatin accessibility.

Although an evident path that bridges DNA methylation editing in promoter and the manipulation of gene repression has not been guaranteed in this manuscript, multiple critical implications have been provided herein, like the list of hypermethylating sensitive promoters, promoter switch in DNA methylation editing, involvement of TETs in differential methylation retention, *et. al.* Nevertheless, it is believed that following revisions are essential to fit Genome Biology's requirement.

1) The simultaneously multiply (thousands) DNA methylation editing relies on a single zinc-finger-DNMT3A fusion protein. Multi-targeting capability is mainly based on the relatively broad specificity of the Sox2 zinc finger. Among the targeted sites, only ~36% encode the 12 bp core sequence, which rises the concern that genome-wide DNA methylation gain may be irrelevant to the limited specificity provided by ZF. More DNA methylation editing profiles and the distribution of ZF-DNMT3A binding peaks data from a few additional ZFs-DNMT3A fusion proteins could solidly support authors' conclusion.

Response: We would like to clarify this observation. Only 36% of the 12 bp core sequence (CCCTCCTCCCC perfect matches) found in the genome overlap with ZF-D3A-wt ChIP-seq peaks. However, among the tested sites (the ZF-D3A-wt ChIP-seq peaks), 75.9% encode the degenerate ZF motif shown in Figure 2a. That is absolutely expected, as all TFs bind degenerate motifs instead of just a core sequence. What this analysis aims to pinpoint is that some potential binding sites of ZF-D3A-wt in the genome remain inaccessible despite encoding a potentially perfect "target" binding sequence. We think this is an interesting observation, since we observe binding of ZF-D3A-wt to non-accessible regions. Similarly, not even pioneer TFs bind to all the possible motifs matches in the genome in physiological conditions. Regarding the remainder of the ~24% of the ZF-D3A-wt ChIP-seq peaks that do not overlap the main motif shown in Figure 2a, we find that they overlap the 2nd or 3rd most

enriched motifs identified in ZF-D3A peaks. The 2nd and 3rd motifs are degenerate versions of the first motif (shorter, or noisier). In total, only 4,405 of the 32,105 peaks lack any of these motifs, which is only 13% of peaks. Since it is known that 3-D structures are preserved by cross-linking, ChIP-seq can carry over non-directly bound genomic sequences, and also is prone to noise, so processes such as these might account for these remaining 13% of peaks. In sum, we demonstrate that the ZF-D3A construct is strongly sequence specific, but it does bind to degenerate motifs instead of only to a unique sequence (as any TF would do). In the revised manuscript we clarify this in the text, so readers do not get confused (page 8, lines 292-299).

Figure 1. Overlaps between ZF-D3A-wt peaks and the top-3 motifs identified by HOMER2.

The reviewer is correct that in the current manuscript we have only used one ZF construct. However, as we mention in the manuscript, another recent study used a different ZF-DNMT3A construct to methylate thousands of regions in the human genome (Broche et al. 2020). Various observations in that study support our conclusions, for instance, the authors found that not all promoters get silenced upon methylation induction, although they briefly touched on that point, and also found that methylation was lost when the ZF was not expressed anymore. Notably, our study goes beyond the findings reported there, as we have generated many new high-quality datasets (e.g. spike-in normalized RNA-seq, TAB-seq, Bisulfite-ChIP-seq, ATAC-seq, WGBS). Furthermore, they did not report many of the novel observations we describe in this manuscript (e.g. differential TSS usage, TF footprinting with ATAC-seq, nucleosome profiling with nucleosome ATAC, non-CpG methylation deposition, hydroxymethylation, co-occurrence of methylation and H3K4me3/RNAPol2). Therefore we believe that our observations are not anecdotal, but likely in common with the use of other ZF proteins. Nonetheless, generating new ZF-DNMT3A cell lines and providing the same level of detail and characterization that we have achieved here would require the generation of an extensive amount of data (e.g. 18x RNA-seq, ChIP-seq for H3K4me3/H3K27me3/H3K9me3, 10x ATAC-

seq, TAB-seq) that so far has taken years to generate, so we believe this is beyond the scope of the current study.

2) Dox withdraw (DoxWD) experiment provides the valuable views to decipher the retention and loss of DNA methylation marks after editing. However, how the DoxWD samples obtained is not clearly displayed in manuscript. I may miss the method detail of this experimental design, thus it will be helpful if authors could confirm that the Dox withdraw was from the cell population 1) GFP+ sorted cells or 2) the bulk cells (both GFP+ and GFP- after 3-day induction). The decrease of GFP signal may come from the withdraw of Dox induction or the lethal cytotoxicity from the global hypermethylation. The latter situation may partially explain the result why 6-day Dox induction achieved lower methylation level than 3-day induction (Line 766).

Response: Apologies for the lack of clarity in the original description of this experiment. To obtain the “Dox withdrawal” samples, we sorted the GFP+ cells after 3 days of Dox induction and then cultured them in Dox-free media for 7 more days. We did this to avoid any potential GFP- cells that could have been in the culture at day 3 from outcompeting the cells that expressed the ZF construct. We now clearly mention this in the Figure 1 legend, to avoid potential confusion by readers.

Regarding the 6-day induction mentioned in line 766, we cultured the cells for 6 days in media with Doxycycline, then sorted the cells after 6-days of Dox induction and isolated the DNA. The global hypermethylation that is induced could potentially have a cytotoxic effect on the GFP+ cells, and result in any GFP- cells proliferating at a higher rate. The only possible workaround for this would have been culturing cells for 3 days in Dox, sorting GFP+ cells, and growing them for 3 more days in Dox. However, cells grow very slowly after the sorting, therefore, going through several rounds of FACS and Dox induction would have made the system highly inefficient, and collecting enough material for subsequent (epi)genomic profiling would have been unfeasible. As it is well established that achieving 100% mCG induction is highly challenging with any epigenome engineering tools, the 3 days 1000 ng/ul Dox induction was deemed the most suitable.

To avoid any confusion for the readers regarding how this was performed, we now clarify this in the methods section “*Lentiviral constructs, MCF-7 cell lines and FACS sorting*”.

3) The intermediate status between Dox and DoxWD (day 7) is critical to figure out the role of TET-mediated demethylation in controlling the rates of methylation loss. An additional time point at day 2 or day 3 post the Dox withdraw will provide the insight regarding correlation between the status of TF binding/chromatin accessibility and the rate of methylation loss.

Response: As per Reviewer 1’s suggestion, we have now generated an intermediate withdrawal stage, 3 days post-FACS, which is day 3 post the Dox withdrawal, and we have performed two replicates of ATAC-seq and WGBS. Consistent with our previous results, the intermediate time point shows greater loss of mCG in the Retain-DMRs compared to the Loss-DMRs (see new Figure 7B). Furthermore, this time-point reveals an intermediate state of chromatin accessibility at Day 3 post Dox induction, showing accessibility values between Dox induction and 7 days withdrawal (the new ATAC-seq data can be seen in the new Figure 7H). Most interestingly, the initial accessibility in distal Retain-DMRs and Loss-DMRs is relatively similar, yet after methylation induction, Retain-DMRs recover more slowly than Loss-DMRs (new Figure 7H). Furthermore, both Day 3 withdrawal datasets have been included in the QC Supplementary Figure S13, showing consistent patterns with previously generated ATAC-seq

and WGBS, showcasing the high reliability of our system (these cell lines have been thawed and re-grown to perform these experiments).

4) The manuscript focuses on the study of the repressive effect from promoter methylation editing. However, only H3K4me3 mark profiles are included. The repressive histone modification marks like H3K9me3 and H3K27me3 are not mentioned in current manuscript.

Response: As per Reviewer 1's suggestion, we have now generated ChIP-seq data for both H3K9me3 and H3K27me3 in NoDox, Dox, and Dox-mutant samples. These data show that while H3K9me3 is not present and therefore is unaffected at DMRs, there seems to be some deposition of H3K27me3 upon DNA methylation induction in some regions (see new Supplementary Figure S6E). We also observe that promoter-DMRs associated with genes that get upregulated upon Dox induction tend to have higher levels of H3K27me3 in the promoter region (Supplementary Figure S6E).

Furthermore, we observe that DMRs that tend to retain DNA methylation for longer show higher H3K27me3 induction (see new Figure 7I), which suggests that these two epigenetic marks reinforce each other on certain susceptible regions (e.g. those with lower basal accessibility). This corroborates previous observations requiring a combination of histone methyltransferases and DNA methylation to obtain more stable methylation retention (Amabile et al. 2016; Nuñez et al. 2021). These observations are also highly consistent with a recent study done with a different ZF-D3A in HEK293 cells (Broche et al. 2020).

We have modified the manuscript accordingly to describe these interesting new observations, and thank Reviewer 1 for their helpful suggestion.

5) Extra discussion regarding the difference of ChIP-seq mapping signals between ZF-D3A-wt and ZF-D3A-mut enrichment (Figure S2B) should be provided. More than a half of peaks are ZF-D3A-wt only is an amazing discovery.

Response: To address this point raised by Reviewer 1, we have now generated a third ChIP-seq replicate for both ZF-D3A-wt and ZF-D3A-mut after Dox induction. In this occasion, the ZF-D3A-mut ChIP-seq data shows a better signal to noise ratio, with a comparable signal to ZF-D3A-wt on shared peaks (see new version of Supplementary Figure 2). Despite the change in signal to noise (% of reads in peaks), the correlation levels with the previous two replicates are very high for both D3A and D3A-mut constructs (Supplementary Figure 13B), highlighting that it is not an artifact or a contamination. Peak calling is very sensitive to ChIP-seq technical variation, and therefore, despite some of the peaks being able to be called exclusively in one sample (or even one replicate), that does not mean the protein-DNA binding enrichment is missing. For example, the "ZF-D3A-wt only peaks" in Supplementary Figure 2B show clear enrichment of signal in the ZF-D3A-mut samples. Despite there potentially being some slight variation in the way these proteins are binding, our data indicates that overall the ZF-D3A-mut largely recapitulates the binding of the ZF-D3A-wt protein. We have now re-written the corresponding paragraph in the manuscript to reflect on these changes (page 7, lines 263-277).

6) Data details or interpretation should not be included in Method section (Line 766 - 771).

Response: As per Reviewer 1's suggestion, we have now removed data details and interpretation from the Methods section.

7) Current typesetting needs further polishing. The full name of abbreviation terms (like "CTD" in Line 559; "mC" in Line 545, et.al.) are not included, and typo errors (like "reconFig. DNA" in line 635, et.al.) should be revised.

Response: We have fixed these typos in the revised version of the manuscript.

Reviewer #2:

This manuscript from de Mendoza et al. describes a fascinating set of -omics experiments where they broadly target a zinc-finger methyltransferase across the genome of MCF7 cells and analyze the resultant effects on DNA methylation, chromatin structure, and expression. There has been much recent interest in epigenome editing as an effective measure of the downstream functional effects of methylation changes as well as a potential therapeutic. However, our understanding of how to control this approach and how the genome constrains this technology is still fairly limited. This work would thus help in the future design of epigenome targeting experiments. The work is quite rigorous, and the positive conclusions (e.g. that sometimes methylation can act as a repressor or activator depending on what TF is affected) are important and well supported. However, there are some broader negative conclusions about methylation-mediated promoter silencing that are not well supported since it's difficult to prove a negative conclusion, especially without substantial more work. For example, it's very hard to sort out the effects at individual promoters when large numbers of genes changed, including potential silencing or activation of other cis-acting factors. Further it's difficult to rigorously assess that the level of induced methylation and the sites methylated were sufficient to induce methylation as would be expected by the standard hypothesis that promoter methylation silences expression. I do wonder how much of what they learned here would transfer to CRISPR-based editing systems. These systems are potentially more impactful since they are cheaper and easier to implement. That said, there is truly an impressive assortment of -omics analysis to support their observations. Many of the major points about how induced methylation may silence or activate gene expression depending on the context are under appreciated and this work establishes that global trends based on these mechanisms are observed.

Major Comments

(1) Some conclusions are overstated, especially concerning how promoter methylation may not silence expression. While they very well may be right, this is hard to prove, and the data presented do not go far enough to support these broad statements. For example:

a. In the abstract: "methylation is often not sufficient to suppress transcription" - this would take a lot more work to show that unsilenced promoters were sufficiently methylated (most were only methylated at < 40%) across the entire promoter or CGI and in multiple contexts. This is not sufficiently tested. Instead, there is a beautiful demonstration that methylation can have a variety of context specific effects (e.g. silencing, activation, repression)

Response: To accommodate Reviewer 2's suggestion we have now changed the title of the paper and the relevant sentence in the abstract, to highlight the context-specific transcriptional responses rather than the lack of repression.

b. In the abstract: "in a process combining active and passive demethylation". - just because it is possibly TET mediated does not mean it's not also cell division dependent, which usually falls under "passive demethylation" rather than active. Better to say something like passive and TET-mediated demethylation. There is a similar problem with the conclusion on page 28 line 625.

Response: This is a good point that TET-mediated oxidation does not necessarily imply "active" demethylation, so we have changed this throughout the manuscript accordingly.

c. In the conclusions, "this work constitutes a broad assessment of the repressive capacity of promoter DNA methylation". Again, this isn't really true, since it's hard without substantial more work to conclude that every promoter was induced sufficiently to test it's repressive capacity. The title uses similar problematic language and in this way it's not really indicative of what the major part of the work is about.

Response: We agree, and we have now also rephrased this sentence in the conclusion as well as the title.

(2) In the TET analysis, how significant is the small amount of hydroxymethylation? It appears this is ~2% from TAB-seq data. Is this found at ~2% across all retain and loss DMRs, or is it much higher at a subset DMRs and near 0 at others at much higher levels? 2% is found at retain and loss DMRs equally, what about non-DMR promoters? I.e. is 2% just a general background rate. Is there evidence that 2% at a promoter is indicative of methylation turnover? Or just a minor inconsequential background rate.

Response: While it is true that the values for TAB-seq are not particularly high when compared to WGBS mC levels, this is an inherent limitation of TAB-seq, which has a high false negative rate. Despite the non-conversion rates (false positives) tending to be negligible (< 0.4%), the protection of hydroxymethylated sites by the glycosylation reaction is rather ineffective, rendering false negative rates ~40% (sites that were originally hydroxymethylated but are read as Ts). Therefore, these relatively low values need to be interpreted according to the limitations of this particular technique. Furthermore, hmC is certainly a transient mark, as the reviewer rightly highlighted above, and hmC is not maintained upon cell division by DNMT1, thus it is not expected that large % of CpGs will display hmC at any given time (moreover when the methylation inductions are not 100%).

In any case, we have now performed a new analysis to demonstrate that the hmC signal that we observe on DMRs is not simply background noise. We took promoters of all transcripts (-2 kb, +200 bp), and classified them into 3 categories: 1) not overlapping a DMR, 2) overlapping a Retain-DMR, and 3) overlapping a Loss-DMR (further filtering them for minimal TAB-seq coverage of 4 and encompassing at least 5 CpGs). This figure is now presented in Supplementary Figure S11E, and it shows that promoters without DMRs show lower levels of hmC than those with DMRs (p-value < 1^{-16} one-sided t-test). In contrast, the difference between promoters overlapping Retain-DMRs or Loss-DMRs is lower (p-value = 0.0001). This pattern is consistent with that of the Distal regulatory sites. To show this, we classified ATAC-seq peaks that do not overlap promoters in the same three categories (no DMR overlap, Retain-DMR, or Loss-DMR overlaps). Again, distal sites that do not overlap a DMR showed lower hmCG levels (p-value < 1^{-16}) than those that do overlap DMRs (Supplementary Figure S11E). The difference between distal Retain-DMR or Loss-DMRs was negligible (p-value =

0.08). It is noteworthy that both Promoters and Distal Regulatory sites are defined according to coordinates that are not dependent on DNA methylation (that is, they are not DMRs). We did this to preserve homogeneity across those that overlap with DMRs with those that do not (for which it would be impossible to define methylation-based coordinates). This coordinate choice implies that for some promoters or distal ATAC-seq peaks, parts of those regions will be methylated and therefore susceptible to get hydroxymethylated, explaining why the no-DMR category is not 0% hmCG.

(3) I was surprised in the motif search where the authors identify potential binding sites (p. 9 and Fig 2B) that they chose a single consensus motif rather than running a FIMO or similar search with the entire position specific weight matrix. Do the results change substantially when you open it up to all potentially bound sequences?

Response: We apologize for the confusion that Figure 2B caused. We used an exact match to the original sequence to which the ZF was designed to bind (i.e. CCCTCCTCCCC) simply to pinpoint that the ZF-D3A is not binding to all the possible exact matches encoded in the human genome (the blue circle in Figure 2B). One could expect that over-expressing a ZF could saturate the binding to all possible sites in the genome, yet this is not what we observe (the overlap in Figure 2B showing ~36% of potential binding sites).

However, we already searched for the consensus binding motif using HOMER2 (an analogous tool to FIMO), using the degenerate motif displayed in Figure 2A. This is the darker red circle found in Figure 2B. In sum, what we wanted to show is the contrast between the original expectation of ZF binding (exact match) and the reality, which is a degenerate motif represented as a PWM. The top sequence motif identified from ZF-D3A-wt peaks is found in 75% of ZF-D3A-wt peaks, yet secondary motifs are also found in up to 90% of peaks (for more information please also see Response 1 to Reviewer 1). In the revised manuscript we clarify this in the text, so readers do not get confused.

(4) On page 30, line 674. Are these different features true of any ZF-D3A construct? Or are they specific to just this one? For example, it seems this construct was designed to a CGI, so if you didn't design to a CGI would the off targets still be at high-CpG density promoters?

Response: Although other ZF-D3A constructs have been shown to show similar patterns (see Broche et al. 2020), it could be that for ZFs designed to bind to completely different sequence contexts (e.g. AT rich sequences) it could display distinct responses. We now discuss this caveat in the manuscript. However, it has been shown that using dCas9-D3A constructs without a guide RNA, or simply using too much dCas9-D3A, also ends up targeting methylation to CpG-rich promoters (Galonska et al. 2018; Pflueger et al. 2018). We think that this is expected in any over-expression system with off-target effects, as CGIs usually exhibit an open-chromatin state (therefore accessible to overexpressed proteins in the nucleus) and they are also typically unmethylated, therefore more likely to gain off-target methylation than regions that were already methylated.

(5) I think there are some caveats to their heterogeneity analysis on p. 17 and 18. First reads are not long enough to consider a full +/-2kb region across a broad promoter window (I think that's what they used to call promoter DMRs). For example, the fully methylated reads could be coming from a section of the promoter whose methylation has no bearing on gene silencing, while a large different section does with intermediate methylation. Second, they

made large numbers of modifications across the genome at different loci, it is possible for some cells to lose repressors or gain activators or alternative promoters and others to not. This could dampen transcriptional readouts as well. Further complicating matters, MCF7 cells are a heterogeneous mix, even before you start hitting them with different constructs.

Response: We agree that there could be still unaccounted heterogeneity on those DMRs. We define the DMRs exclusively based on statistical thresholds (using dmrseq), yet we define promoter-DMRs by overlapping DMR coordinates with promoters (defined as -2kb/+200bp), so these DMRs are not necessarily encompassing the whole arbitrarily defined promoter. Furthermore, the reads that we analyze in the per-read methylation plot (Figure 4E) only come from the DMR, not the full promoter length. Please see the schematic below, where we show that only the reads that overlapped >80% with the DMR were selected, while discarding the reads (shown with a red cross) that were surrounding it (even if they were in the arbitrarily defined “promoter” regions):

Nonetheless, it is true that positional information within the DMR might have more or less weight on the transcriptional output, which could be different for every gene/promoter, and also that complex secondary effects could drive high over-expression in a reduced subset of cells, which would be read-out as bulk-level upregulation. We now discuss this potential caveat in the manuscript as quoted below (page 13, lines 586-587), yet we believe that this analysis is important. It shows that non-repressed promoter-DMRs at least do not have more inherent per-read methylation heterogeneity than repressed DMRs.

(6) In the discussion there is mention that DNMT3A + 3L + KRAB was shown to be sufficient for longer term silencing. I also thought there has been some discussion in the literature of the need to hit loci harder (e.g. multiple DOX inductions or with SunTag system) and that these could stabilize the methylation even without KRAB recruitment.

Response: Despite some early reports based on a handful of loci reported that targeting D3A to some promoters could lead to long-term silencing, the consensus in the field is that usually that is not sufficient for long-term methylation retention, see for instance this recent review (Nakamura et al. 2021) or this article (O'Geen et al. 2022). In our experience using dCas9+SunTag-D3A, we did not achieve long term induction of methylation at any loci (Pflueger et al. 2018). Other publications based on the SunTag system have not profiled long time series post methylation induction. This is critical, since we now show in Figure 7B that some DMRs still show relatively high mC levels after 3 days post Dox treatment, but this methylation is essentially fully lost on Day 7 withdrawal. Therefore, unless we are missing

some relevant literature on this topic, we think that we are presenting a balanced view in the field.

(7) Since CRISPR-based epigenome editing is more common (since it's substantially easier and cheaper) than zinc-finger based systems, it would be good if the discussion could talk more about which of their findings will likely hold up in these other systems based on current literature, and which are potentially specific to the zinc finger system.

Response: In the revised manuscript we have now included some discussion of possible specificities of the ZF platform that would not be applicable to CRISPR-based systems. The most important is likely the amount of methylation induction done in parallel, which would be highly ineffective in a CRISPR system as it would require a very high number of guide RNAs in parallel in all cells, which has not been achieved before. Regardless of the targeting system, we anticipate that many of the complex context-specific responses to DNA methylation induction that we describe here will be largely applicable to other systems.

(8) There is a fundamental limitation to this work in that it is only performed in a single cell line. The authors acknowledge this limitation and personally I find this work quite compelling. However, my guess is critics will use this as a dismissive comment. This is merely a comment to the authors. I certainly do not expect them to repeat their findings in another line at this stage. But I did want to mention it in case it is helpful. One possible way to help deal with such critiques would be when they highlight how different observations are supported in the literature in the results, they could add a quick comment the cell line or context.

Response: We thank Reviewer 2 for their suggestion, and agree that the fact that this work is done on only one cell line (MCF-7) could cause some readers to question the generalizability of the observations and conclusions. In the manuscript we have mentioned this limitation in the conclusions section. Yet, as noted above, many similar observations coming from various different DNA methylation editing systems and manipulated cell lines agree with many of our findings, e.g. see (Pflueger et al. 2018; Galonska et al. 2018; Broche et al. 2020). As per Reviewer 2's suggestion, we now highlight the cell type used in other studies to make this point more evident to the readers.

Minor Comments

(a) p. 3, line 72 "5' position" should be "5 position".

Response: We have fixed this typo.

(b) p. 9 line 231 states that the data indicates "that the ZF-D3A binding capabilities are similar to those of a pioneer TF." And then later on p. 11 they seem to conclude the opposite. On p. 9 they might be better of stating that this data suggests that ZF-D3A could act as a pioneering factor, so we decided to test this using the following analysis. Or something like that to avoid confusion of conflicting conclusion-like statements.

Response: We have modified this sentence to avoid this interpretation.

(c) p. 19 line 422, comments about Sox2 lacking methyl-sensitivity is missing a reference.

Response: What we meant is that Sox2 binding methyl-sensitivity is not known, not that it lacks methyl-sensitivity. Moreover, it would be unexpected to find that it is methyl-sensitive, as the consensus motif for Sox2 lacks any potential CpG. We have now re-written this sentence to avoid confusion.

(d) p. 28 line 635, reconFig. Should be reconfigure.

Response: We have fixed this typo.

(e) The wording on p. 30 line 678 seems off. Normally nucleosomes facilitate D3A-dependent methylation deposition. The "even using" construct implies that for both their fusion and in normal function nucleosomes impede methylation. I think the wording just needs to be adjusted a little.

Response: We have re-written the sentence to avoid this potential interpretation.

Reviewer #3:

I've read this updated manuscript with great interest. It is a vastly improved version of the manuscript that was on bioRxiv already since 2017. I have to say that this is excellent work, highly important for the field and for understanding the epigenetic regulation and the role DNA methylation plays. The study is very well designed and controlled, while taking into consideration many alternative explanations it clearly shows that DNA methylation is not sufficient to provide gene repression and that additional factors also play a role. Whereas for the initial bioRxiv manuscript, I had doubts to some parts and the data could be explained otherwise, it seems that the authors managed to very comprehensively address them in this submitted version.

Response: We would like to thank the reviewer for the positive evaluation of this manuscript and for the recognition of the substantial efforts we have done to improve this work upon the original pre-print.

Second round of review

Reviewer 1

Generally, the revised version has fully addressed my concerns regarding the representative of ZFN sequence, intermediate status after Dox withdrawal, repressive histone marks, and ZF-D3A binding profiles using ChIP-seq. Although few limitations like single cell line utility remain, generating extra cell lines with comparable data is, for sure, beyond the scope of this study, which may also be a consensus from me and reviewer #2. I believe the highlights of this manuscript are the insight into the complexity of manipulating expression at the epi-level and potential help in the future design for epi-editing. Thus, the updated version deserves further consideration in Genome Biology.

Reviewer 2

I appreciate the time and effort the authors spent addressing my prior comments. This is an excellent work.